# Interannual variability of the gravity wave drag - vertical coupling and possible climate links

Petr Sacha[1,2], Jiri Miksovsky[1], and Petr Pisoft[1]

[1]Department of Atmospheric Physics, Faculty of Mathematics and Physics, Charles University, V Holesovickach 2, 180 00 Prague 8, Czech Republic
[2]Faculty of Sciences, Universidade de Vigo, Ourense, Spain

*Correspondence to:* Petr Sacha (petr.sacha@mff.cuni.cz)

**Abstract.** Gravity wave drag (GWD) is an important driver of the middle atmospheric dynamics. However, there are almost no observational constraints on its strength and distribution (especially horizontal). In this study we analyze orographic GWD (OGWD) output from Canadian Middle Atmosphere Model simulation with specified dynamics (CMAM-sd) to illustrate an interannual variability of the OGWD distribution at particular pressure levels in the stratosphere and its relation to major climate oscillations. We have found significant changes of the OGWD distribution and strength depending on the phase of the North Atlantic oscillation (NAO), Quasi Biennial oscillation (QBO) and El Niño-Southern oscillation (ENSO). The OGWD variability is shown to be induced by lower tropospheric wind variations in a large part and there is also significant variability detected in near surface momentum fluxes. We argue that the orographic gravity waves (OGWs) and GWs in general can be a quick mediator of the tropospheric variability into the stratosphere as the modifications of the OGWD distribution can result in different impacts on the stratospheric dynamics during different phases of the studied climate oscillations.

# 1 Introduction

Although the internal gravity wave (GW) sourcing (e.g. adjustment processes, Plougonven and Zhang, 2014), propagation and breaking is governed to some extent by processes in the stratosphere, there is a significant portion of the GW spectra created in the troposphere (mostly orography and convection, Alexander et al., 2009). The highest amplitude upward propagating modes can break already in the troposphere and lower or middle stratosphere (Fritts et al., 2016). Model experiments with gravity wave drag (GWD) parameterization showed that the orographic GWD in the lower stratosphere can significantly affect the development of the sudden stratosperic warming (SSW) (Pawson, 1997; Lawrence, 1997; Šácha et al., 2016; White et al., 2017, 2018) and the large-scale flow in the lower stratosphere and troposphere in general (McFarlane, 1987; Alexander and Shepherd, 2010; Sandu et al., 2016; Šácha et al., 2016; White et al., 2017). In the global climate models, non-orographic GWs are usually considered to be breaking higher above starting at the upper stratosphere (Scinocca, 2003). It is well recognized that there is a need for continued and additional research efforts on stratospheric dynamics (Añel, 2016) as complex understanding and unbiased modelling of stratospheric conditions is vital for climate research (Manzini et al., 2015; Calvo et al., 2015).

From sensitivity simulations with a mechanistic model, Šácha et al. (2016) demonstrated dynamical impact of the artificially enhanced GWD in the stratosphere and most importantly significant impact of the spatial GWD distribution. This can open new horizons for research of teleconnections between tropospheric (e.g. El Niño-Southern Oscillation, North Atlantic Oscillation, Pacific Decadal Oscillation) and stratospheric (e.g. polar vortex stability) phenomena taking into account that the tropospheric variability can affect the distribution of GW sources and therefore the GWD distribution (and strength) in the stratosphere. This is also the main hypothesis that we investigate in this study. It is not possible to compute the GWD from current satellite observations alone (Alexander and Sato, 2015; Geller et al., 2013). Only by employing harsh approximation and neglecting observational filter effects, Ern et al. (2011) gave a methodology to estimate absolute values of a "potential acceleration" caused by GWs. Some information can be derived also using ray-tracing simulations (Kalisch et al., 2014). However numerical simulations remain the major source of the GWD variability global description. This is also the reason why we study the interannual variability of the GWD using output from Canadian Middle Atmosphere Model with specified dynamics (CMAM-sd) in this paper. Although the orographic GW parameterization schemes present a severe simplification of the reality (e.g. assuming vertical propagation only, Kalisch et al., 2014), it is the only available source providing three-dimensional decadal long information on the GWD that is necessary to test the hypothesis of connection between the climate oscillations and GWD distribution. To our knowledge the interannual variability of GW model parameterization outputs has not been studied before. The study is structured as follows. The next section introduces the model, SD simulation and the orographic GWD (OGWD) parameterization scheme together with statistical methods used in our study. Second section is dedicated to the OGWD analysis, assessing realism of its climatology first. This is followed by an interannual variability analysis where significant differences in distribution of OGWD depending on Southern Oscillation (SO), North Atlantic Oscillation (NAO) and Quasi Biennial Oscillation (QBO) are illustrated. In the third section we examine the correspondence of OGWD to tropospheric conditions and analyze the variability of OGW momentum fluxes at the 850 hPa level. Finally, summary of results and discussion of uncertainties and implications of our paper are given.

## 2 Methodology

### 2.1 CMAM-SD and its GWD parameterizations

Canadian Middle Atmosphere Model (CMAM) chemistry climate model with 71 levels up to about 100 km with variable vertical resolution and a triangular spectral truncation of T47, corresponding to a 3.75° horizontal grid has been used for
producing the specified dynamics (SD) simulation of the time period between 1979 and 2010. Up to 1 hPa the horizontal winds and temperatures are nudged to the 6-hourly horizontal winds and temperatures from ERA-Interim (Dee et al., 2011), as described in more detail in McLandress et al. (2013). Due to this nudging, CMAM not only realistically reproduces the climate characteristics of real atmosphere, but also follows its historical trajectory in a deterministic sense. This also applies to the activity of internal climate variability modes and their spatial response patterns, as illustrated by the samples in Figures S1
and S2 in the Supplement.

    Orographic GWD (OGWD) is parameterized using the scheme of Scinocca et al. (2000). This OGWD scheme employs two vertically propagating zero-phase-speed GWs to transport the horizontal momentum to the left and right of the resolved horizontal velocity vector at the launch layer, which extends from the surface to the height of the subgrid topography and the static stability. Functional dependence is on the near-surface wind speed, relative orientation of the subgrid topography
(determining the orientation of the GW momentum flux) and the static stability in the source region. There are also two dimensionless parameters in the OGWD scheme allowing to arbitrarily control the total value of launch momentum and the vertical flux of horizontal momentum (more detail in McLandress et al., 2013) influencing also indirectly the breaking level. The setting used in CMAM-sd has been tuned for polar-ozone chemistry studies in CMAM since it produces reasonable zonal-mean zonal winds and polar temperatures in the winter lower stratosphere (Scinocca et al., 2008). As the parameterized
orographic GWs propagate upward they are subject to both critical level filtering and nonlinear saturation (using a convective instability threshold), where the functional dependence is on the resolved horizontal wind speed and direction and static stability on the place (refer to Scinocca et al. (2000) for exact description).

    The CMAM non-orographic GW parametrization scheme (Scinocca et al., 2008) is based on launching a globally uniform isotropic non-orographic GW spectrum in four cardinal horizontal directions at approximately 125 hPa. The aim is to
produce reasonable seasonal evolution of the zonal mean zonal temperature and winds in the mesosphere. The zonal and meridional asymmetry stems from propagation effects only. From those reasons it is clear that the resulting non-orographic GWD (NOGWD) is not suitable for our analysis.

### 2.2 MLR and other statistical methods

The specific GW responses to the changes in model atmospheric circulation can be quite non-trivial, as their functional de-
pendence on the background quantities is nonlinear and their extraction and quantification requires application of statistical methods able to separate the effects of multiple simultaneously acting factors. Here, the association between OGWD and selected prominent climate variability modes has been investigated through multiple linear regression (MLR), using scalar indices of the North Atlantic Oscillation (NAO, defined as a normalized pressure difference between Reykjavik, Island and

Gibraltar), Southern Oscillation (SO, defined as normalized pressure difference between Darwin, Australia and Tahiti) and Quasi-Biennial Oscillation (QBO, defined as zonal average of equatorial zonal wind at 30 hPa) as explanatory variables, along with descriptors of solar forcing (total solar irradiance), volcanic forcing (global mean stratospheric volcanic aerosol optical depth) and linear approximation of the long-term trend component. The time series of the respective indices were used in the form available from the KNMI Climate explorer database (https://climexp.knmi.nl). Additional experiments have also been carried out to investigate the effects of internal climate variability modes with dominant decadal and multi-decadal components: Pacific Decadal Oscillation - PDO and Atlantic Multidecadal Oscillation - AMO. However, due to their largely statistically non-significant influence on GWD, as well as aliasing with other predictors (particularly the Southern Oscillation index), only results obtained without considering PDO and AMO are presented here. Statistical significance of the regression coefficients has been estimated by moving-block bootstrap, with the block size chosen to accommodate for the autocorrelated structures in the regression residuals. MLR has also been used to assess the associations between GW effects and local circulation (characterized by geopotential height or wind speed at various pressure levels); step-wise version of linear regression was used for some of these analysis setups, to identify the predictors most relevant to the OGWD output. Due to the distinct annual cycle of the activity of the orographic GWs (with their strongest manifestations typically observed during the cold part of the year), seasonal specifics need to be considered in the attribution analysis. While sub-seasonal setup (such as analysis carried out separately for individual months of the year) would be desirable, it would be difficult to achieve because of the relative shortness (mere 32 years) of the time series analyzed here and the resulting limited amount of independent samples. For this reason, separation into traditionally defined climatological seasons was used instead.

## 3   Results

GW influence on the stratospheric circulation is often estimated and confronted with forcing from resolved waves on the basis of zonal means (see e.g., Albers and Birner, 2014). However, as we show in the first section of results, the CMAM-sd OGWD climatological horizontal distribution at 100, 50, 30 and 10 hPa is highly zonally asymmetric and OGWD tends to be distributed in local hotspots. The different dynamical effect of hotspots instead of zonally symmetric forces have been already shown numerically by Šácha et al. (2016). Results in the next sections illustrate that the studied atmospheric phenomena are connected with different OGWD distribution and thus with potentially different impact on the stratospheric dynamics.

Geller et al. (2013) made a first formal comparison between GW momentum fluxes from models and observations concluding that the geographical distribution of the fluxes from models and observations compare reasonably well, except for certain features connected mainly to non-orographic GWs. We are interested mainly in the geographical distribution and so we simply confront the CMAM-sd OGWD hotspots with observed GW activity (not only momentum flux) hotspots on selected pressure levels in the stratosphere. The CMAM-sd momentum flux climatologies are shown in the Supplement.

## 3.1 CMAM-sd GWD climatology

First, we examine if the orographic GW parameterization scheme from CMAM-sd distributes the OGWD realistically. Fig. 1 shows the OGWD climatology at 100, 50, 30 and 10 hPa levels. The 100 hPa level is traditionally below the level taken into account in the GW analyses from satellite observations (e.g., Alexander et al., 2010; Šácha et al., 2015; Wright et al., 2016). At this level, in the DJF season, the OGWD is dominated by Himalayan hotspot, which has not received significant attention in observational analyses yet (probably due to its emergence at rather lower levels). However, enhanced momentum fluxes have already been observed in this region e.g. by Wright et al. (2016). Another hotspot emerging in the Northern Hemisphere (NH) is connected with the Rocky Mountains. These hotspots are not visible at the higher levels. In the Southern Hemisphere (SH), during southern summer conditions, we see comparable magnitudes of OGWD (up to 20 m/s/day) as for the NH connected with the southern tip of Andes, Tasmanian Island and New Zealand. Those high OGWD values in the summer hemisphere vanish at higher levels, which is in line with Baumgaertner and McDonald (2007), who attributed the small amount of summertime potential energy to lower level critical filtering.

In the JJA season at 100 hPa, there is no dominant hotspot in the NH, while the SH OGWD distribution is dominated by the hotspot connected to Andes. At 50 hPa in the DJF, there is a dominating hotspot in the region of eastern Asia corresponding to the Eastern Asia/Northern Pacific (EANP) hotspot observed by Šácha et al. (2015) or referred to as Mongolian orography in White et al. (2017, 2018). In the SH in the JJA season Andes are dominant, but note that the OGWD magnitude is smaller than for the EANP in the DJF season. Interestingly, at 30 hPa we see dominance of the same hotspots as at 50 hPa but with smaller magnitude of OGWD. In the SH in the JJA, the OGWD around Drake Passage and Antarctic Peninsula (de la Torre et al., 2012; Hoffmann et al., 2013, 2016; Hindley et al., 2015; Wright et al., 2016) begins to gain strength. At 10 hPa, in the NH in DJF, the Scandinavian hotspot starts to be dominant (John and Kumar, 2012; Hoffmann et al., 2013). In the SH in JJA southern Andes (de la Torre et al., 2012; Hoffmann et al., 2013), Drake Passage and Antarctic Peninsula hotspots dominate. Interestingly, we can see also moderately strong OGWD (Fig. 1, 50hPa, DJF) over small remote islands in the SH (Alexander and Grimsdell, 2013; Hoffmann et al., 2013).We conclude that the OGWD distribution from CMAM-sd gives sufficiently realistic distribution of the OGWD for our analysis, given the assumptions employed in the parameterization and the lack of direct observational information on the OGWD and GWD in general.

Fig. 2 gives an illustration of how much the OGWD is changing on the interannual scale. Note that the arrows do not show the drag direction, but illustrate a ratio of the meridional and zonal standard deviation (both always positive). We see that at 10 hPa large OGWD variations correspond to Scandinavia, central Asia and Greenland in the NH and southern tip of Andes together with the region of Antarctic Peninsula in the SH winter. Standard deviation values reach to 20 m/s/day in both hemispheres with prevalence of the zonal OGWD component (except the Antarctic Peninsula). The OGWD variability at the 30 hPa level is dominated by the EA/NP and Scandinavian hotspots with maximum values of standard deviation below 5 m/s/day. This magnitude is reached only in the Antarctic Peninsula region in JJA in the SH. In the NH, meridional component has relatively lower variability than at 10 hPa.

At 50 hPa in the NH winter, we see the largest OGWD variations in the EA/NP hotspot and surprisingly large values also locally in the SH in southern Andes. This is also the only region with pronounced variation of OGWD in the SH winter. Rocky Mountains and esp. Himalayas and southern Andes with standard deviation values around 5 m/s/day dominate the 100 hPa level in DJF. At 100 hPa, the relative contribution of the meridional OGWD component variability is bigger than at 50 and 30 hPa. In SH in DJF and JJA, the variability of Andes dominates.

Generally, the OGWD varies interannualy about a half of the climatological OGWD magnitude (even reaching it at 10 hPa), with respective hotspots dominating the variability at the particular pressure levels of their climatological influence.

## 3.2 MLR results

Responses of the OGWD to the phase of major internal climate oscillations are shown at the 50, 30, 10 hPa levels. In the NH in DJF, by far, the variability connected with NAO dominates (Fig. 3, middle panel) in the sense that it is distributed across the whole hemisphere with many significant regions and responses up to 5m/s/day. As could be expected from the NAO definition, it is mostly pronounced in regions surrounding Northern Atlantic. Note especially that at all analyzed isobaric levels there is a dipole like structure between Greenland and Scandinavia together with coastal areas in other places in Western Europe. That indicates that during the positive NAO phase the GW activity supresses the eastward wind above Greenland and enhances it above Western Europe while the oposite is true for the negative phase. Similar dipole can be found at the western coast of Northern America, but only at the 50 hPa level. At higher levels, the signal above Alaska is more pronounced. The NAO signal is also pronounced in north-eastern America, central Asia and partly in the EA/NP region (at 50 and 30 hPa levels) and in the northern Asia for the 10 hPa level. There is also a significant signal exceeding 2 m/s/day in northern Africa for the 50 hPa level. The SO signal in DJF season in the NH is mostly pronounced at 50 and 10 hPa level. At 50 hPa it constitutes a ring of significant OGWD responses higher than 2m/s/day whereas at 10 hPa level the signal in the north-eastern America, Turkey, Iran and Caucasus region dominates. At 50 hPa there is also a strong localized signal in the southern tip of South America. The QBO signal in DJF is mostly pronounced in the central Asia in the NH and southern Andes together with Antarctic Peninsula at 50 and 30 hPa in the SH.

During austral winter (JJA, Fig. 4), the biggest signal found in the OGWD belongs to SO with domination of Antarctica at the 50 hPa level. At higher levels there is a dipole like feature between Antarctica and southern tip of Andes. There is also a strong (more than 2 m/s/day) significant signal connected with the QBO at the 50 hPa level over Andes. Somewhat surprisingly we can find also significant NAO signal (cca 1 m/s/day) around southern Australia and New Zealand at 50 hPa. Results of the regression of solar activity and volcanic forcing were not shown, because they gain mostly insignificant OGWD signal. Only at 50 hPa, there is a weak (up to 1 m/s/day) significant solar signal in north-eastern America and Antarctica in their respective winter periods.

To illustrate that it is necessary to consider geographical distribution for analysis of the interannual variability of OGWD, we show the MLR results also for zonal means of OGWD (shown for DJF only). For the zonal OGWD component (Fig. 5) we can see that there is only a weak positive significant NAO signal at all levels and a very small positive significant SO signal at 50 hPa between 20-30°N corresponding with the belt described in the discussion of the Fig. 3. The magnitude of the detected

signal is lower than 1m/s/day everywhere. For the QBO and also for the meridional component (Fig. 6) of all indices the signal is not significantly positive or negative or is lower than 0.1m/s/day almost anywhere. Similar holds also for the JJA season (not shown).

General finding of the above presented results is that the OGWD varies locally by a few m/s/day depending on the phase of the climate indices and also that the geographical variation of hotspots can vary from a phase to phase. The analysis points also to an important finding that the significant signal connected to the climate oscillations diminish in case of the traditional zonal mean approach.

## 3.3  Explanatory factors

The above presented results alone cannot confirm our hypothesis on the tropospheric variability transfer to the stratosphere by altering the GW activity and its distribution because the MLR results do not illustrate the causality of the problem considered. It can be argued that the OGWD variability results are caused simply by the variations in the stratosphere or upper troposphere (e.g., jet shift, meandering due to anomalous planetary waves (PW) activity etc.) possibly leading to Doppler shifting effects or variations of critical lines for the orographic GW propagation (e.g., a role of Aleutian High occurrence for the EA/NP hotspot - Pisoft et al., 2018). The modulation of GWs by PWs receives a big attention in the scientific community (e.g., Cullens et al., 2015) and considering this causality mechanism the dynamical influence of the OGWD variations would be of secondary importance only. Therefore, in this subsection we analyze daily data of wind direction and speed (influence of another OGWD parametrization variable - a stability - was not diagnosed) to show that at least a part of the OGWD variability is directly influenced by the variability at the surface or in the lower troposphere.

Figs. 7, 8 and 9 present analysis of daily data aimed at estimating how much of the OGWD variability at a given level can be explained by 850 hPa wind variance. At 50 hPa, we can see that the link between the lower tropospheric winds and OGWD is strongly expressed in a belt in the mid-latitudes and tropics of the NH. The fraction of variance explained is maximal and also the geographical distribution is very similar for the links between zonal wind/zonal OGWD component and meridional wind/meridional OGWD component. In the regions with significant orography and particularly in the region of the EA/NP hotspot (which dominates the OGWD field at 50 hPa in the NH) the majority of OGWD variance is explained by lower tropospheric winds.

Interesting pattern can be seen in the SH around Andes, where the maximum of the OGWD variance explained is located up and down-wind from Andes. Also interestingly, at 50 hPa, in the southern Andes/Antarctic Peninsula region, larger fraction of the meridional OGWD component variance is explained by surface conditions than for the zonal component. Otherwise, the fraction of the OGWD variability explained in the Australian/New Zealand hotspot (connected in previous analyses mainly with the NAO signal) is about two fifths of the total variance.

At the 30hPa level the fraction of variance explained is lower - around one third of the variance in eastern Asia, and locally in northern Atlantic coastal regions and in the SH. In the eastern Asia region this is due to the stratospheric background affecting the critical line occurrence and propagation of the GWs between 50 and 30 hPa. Interestingly, for the meridiodal OGWD component the fraction of variance explained is slightly higher. At the 10 hPa level, there is a single maximum of explained

total variance (around one third) in Scandinavia. Similar amount of variance is also explained by 850hPa winds for Iceland, but for zonal OGWD component only.

Another approach allowing to assess the variability of the orographic GW sourcing is to analyze the 850 hPa orographic GW fluxes as a proxy and apply the MLR method. However, with this method it is not possible to link the results directly with the variability of the OGWD because processes like the Doppler shifting of amplitudes or critical line variations can alter the resulting OGWD significantly. Also note, that this analysis is made on monthly data and the comparability with previous analysis is limited.

In Fig. 10 we see that in the NH in DJF there is a strong signal in Greenland and western Europe connected with the NAO and equally strong signal in GW sourcing variability in central Asia (Himalayas), Greenland, Iceland and Svalbard connected with the QBO. The SO signal is largely insignificant in the NH, but in the SH in JJA it is strongest pronounced mainly in the southern tip of Andes and Antarctic Peninsula. In the SH in JJA there can be found also some regions of significant signal in GW sourcing variability connected with the NAO (Andes, Australia and New Zealand) and QBO (Antarctica).

Although the strong QBO signal may be surprising, the QBO phase exhibits a distinct and in some regions statistically significant influence on the lower tropospheric winds in CMAM-sd (not shown). The influence of the QBO on the surface meteorological conditions has been pointed out in the literature in detail before (e.g., Marshall and Scaife, 2009; Hansen et al., 2016).

## 4  Summary and discussion

The presented study introduces an analysis of interannual variability of the CMAM-sd OGWD at particular pressure levels in the stratosphere. Building on the results of Šácha et al. (2016), the aim of our paper has been to evaluate if the tropospheric variability can affect the OGWD distribution in the stratosphere.

In the first section we show the simulated climatological OGWD distribution at 100, 50, 30 and 10 hPa levels and estimate its interannual variability to be about half of the climatological OGWD value at the major hotspots. The main conclusion of this part is that the distribution can be regarded as reasonably realistic because the main GW activity hotspots are detected similarly as they are described in the GW observing literature (also considering the practically missing observational constraints on the OGWD in general). In the second section, results of the MLR analysis of monthly OGWD data are presented showing significant NAO, SO and QBO signal of a few (up to 5) m/s/day in the OGWD at 50, 30 and 10 hPa. Depending on a phase of the climate oscillations, OGWD values in the hotspot regions and also the distribution of OGWD hotspots vary interannualy on the selected pressure levels. However, in case of the traditional zonal mean analysis, the detected signal is small and mostly insignificant. In the last part we demonstrate that a large fraction (over hotspots like EANP) of the described OGWD variance can be linked to the variance of 850 hPa winds. Also we find significant NAO, SO, QBO signal in the orographic GW momentum fluxes at 850 hPa suggesting different orographic GW sourcing in a model depending on a phase of those phenomena.

All of the results support for CMAM-sd simulation the original hypothesis of the tropospheric variability transfer into the stratosphere via OGWD variability. The suggested mechanism depicts a simplified picture not taking into account the inner variability of the stratosphere, PW propagation or mutual interactions between the troposphere and stratosphere. On the other hand, it has to be noted that the GWs are arguably the fastest way for communication of information in the vertical (apart from the acoustic and acoustic-gravity waves with effects much higher in the middle and in the upper atmosphere). Therefore the tropospheric information can be quickly mediated into the stratosphere and the OGWD variability can be directly influenced by the variability at the surface or in the lower troposphere. During propagation and in the stratosphere, those fast and GW mediated tropospheric contributions interact nonlinearly with the stratospheric processes (Doppler shifting, critical level variations). However, it makes little sense to look for the causality between GWs and PWs (background field for GWs) when only the steady state (monthly data) is considered.

There is also a factor of longitudinal variability of the OGWD (and GWD in general). For the PW breaking there is almost no information in the literature about the geometry and longitudinal variability of the imposed drag force. But for the GWs, it has been shown in Šácha et al. (2016) that localized forces can lead to dynamical responses different from the reactions to a zonally averaged forcing. Although the gravity waves are a small-scale phenomenon, they are often organized in large-scale hotspots constituting a large-scale forcing. We argue that incorporating those effects into related analyses can open new horizons for research of teleconnections between tropospheric (e.g. SO, NAO or PDO) and stratospheric (e.g. polar vortex stability) phenomena. The magnitude of the OGWD variations reaching to a few m/s/day locally can significantly affect the stratospheric dynamics. It was shown by Šácha et al. (2016) that the injection of a localized versus zonally symmetric GWD of 10 m/s/day can lead to wind speed differences of an order of ten m/s at corresponding vertical levels. For the residual circulation and Elliassen-Palm flux the localized GW forcing of this magnitude induced differences ranging up to 50% of their climatological values in the Middle and upper atmosphere mechanistic model (Pogoreltsev et al., 2007) used for the study Šácha et al. (2016).

Our analysis relies on parameterized processes and thus the results can be highly model dependent considering that other models use different OGWD parameterizations than CMAM. The NOGWD in CMAM-sd at the vertical domain of our analysis is clearly underestimated compared to the current consensus on the GW distribution and impacts on the stratosphere (e.g., Hoffmann et al., 2013; Holt et al., 2017; Polichtchouk et al., 2007). It would be highly interesting to look at NOGWD variations connected with variability of jets, fronts etc. in future research. For the real atmosphere our results strongly suggest that GWs can play a much bigger and different role in the troposphere-stratosphere coupling and in shaping the stratospheric dynamics than is currently acknowledged. However, at the current stage it is impossible to evaluate actual details of the connection between the climate oscillations (tropospheric variability) and OGWD changes. This is partly due to the nudging procedure, which prevents us to analyze the GWD impact on the circulation, because this impact is weakened by the relaxation towards ERA-I data. However, as we have only analyzed the OGWD interannual variability, the nudging is very advantageous for us (compared to a free running model), since the distribution of gravity wave momentum fluxes in CMAM-sd can resemble the distribution of real fluxes (McLandress et al., 2013).

From a methodological point of view we must also note that GWs and their effects are handicapped by the use of monthly mean data because the GWs are very intermittent in the atmosphere (e.g., Hertzog et al., 2012; Wright and Gille, 2013) and also in CMAM the OGWD shows large daily (and shorter, not shown) variability. Therefore the monthly mean values can be hiding e.g. one order stronger intermittent drag values. During the analysis, there were also indications of noteworthy deviations from linear behavior in some regions encouraging future transition to nonlinear regression techniques.

In our future work, we aim to separate and estimate dynamical impacts of the different OGWD distributions belonging to respective phases of the NAO, SO, QBO by producing sensitivity simulations with a mechanistic model with prescribed OGWD values and distribution according to MLR results from CMAM.

*Data availability.* CMAM outputs are available upon registration at the website of the Canadian Centre for Climate Modelling and Analysis http://climate-modelling.canada.ca/climatemodeldata/cmam.

*Competing interests.* The authors declare that they have no conflicts of interest.

*Acknowledgements.* The presented work would not be possible without the utilized CMAM datasets from the Canadian Centre for Climate Modelling and Analysis. This study was supported by GA CR under grant no. 16-01562J and 18-01625S. In later stages of the manuscript preparation, Petr Sacha was supported by the Government of Spain under the grant no. CGL2015-71575-P. Petr Sacha thanks J. Añel and L. de la Torre for fruitful discussions. We thank our colleague Ales Kuchar for his assistance with data preparation. We also thank the two anonymous reviewers whose insightful and constructive comments helped to improve and clarify this manuscript.

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

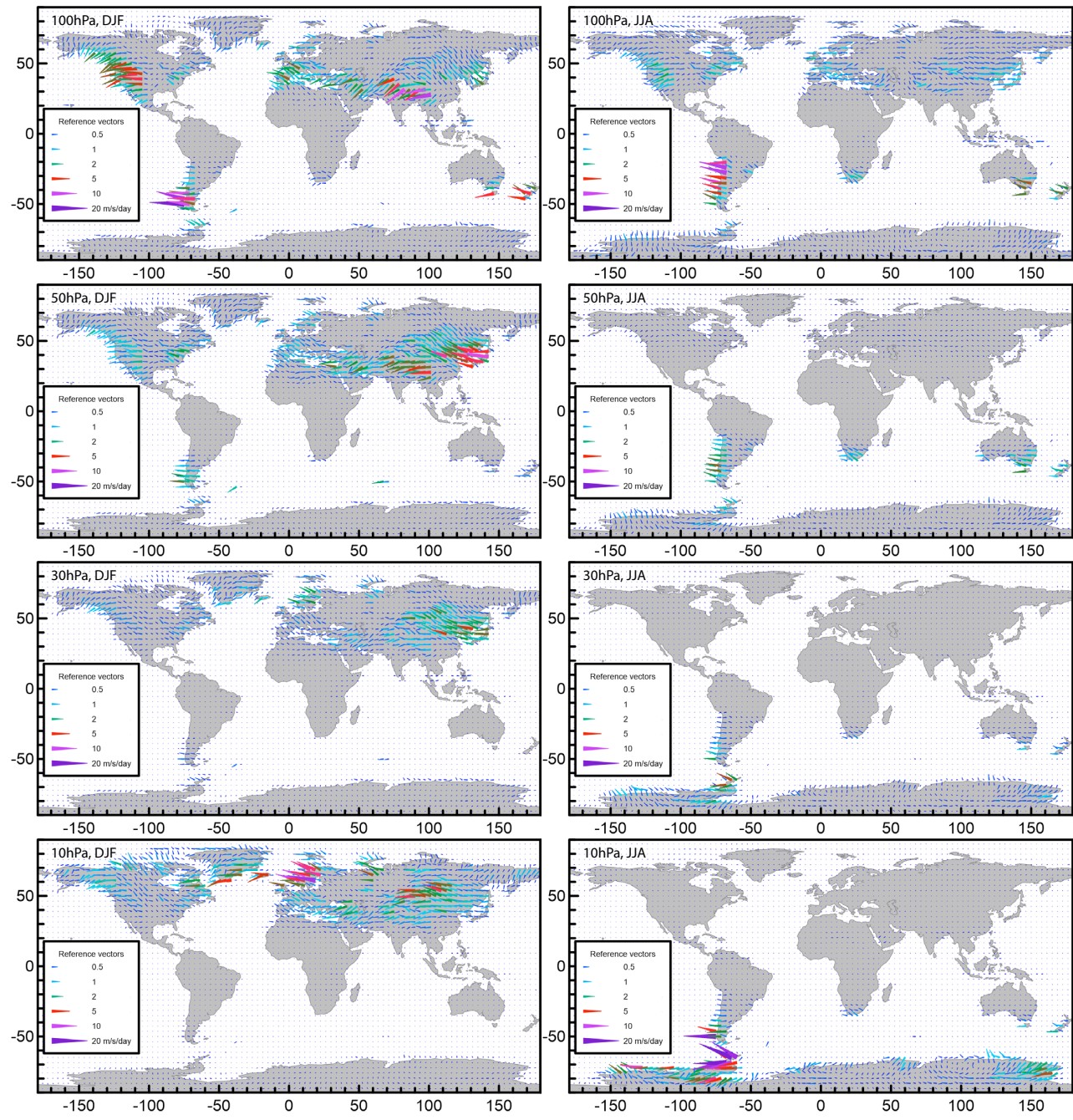

**Figure 1.** Mean seasonal wind tendency due to OGWs [m/s/day] at the 100 hPa (top), 50 hPa (upper middle), 30 hPa (lower middle) and 10 hPa (bottom) level, during DJF (left) and JJA (right) seasons.

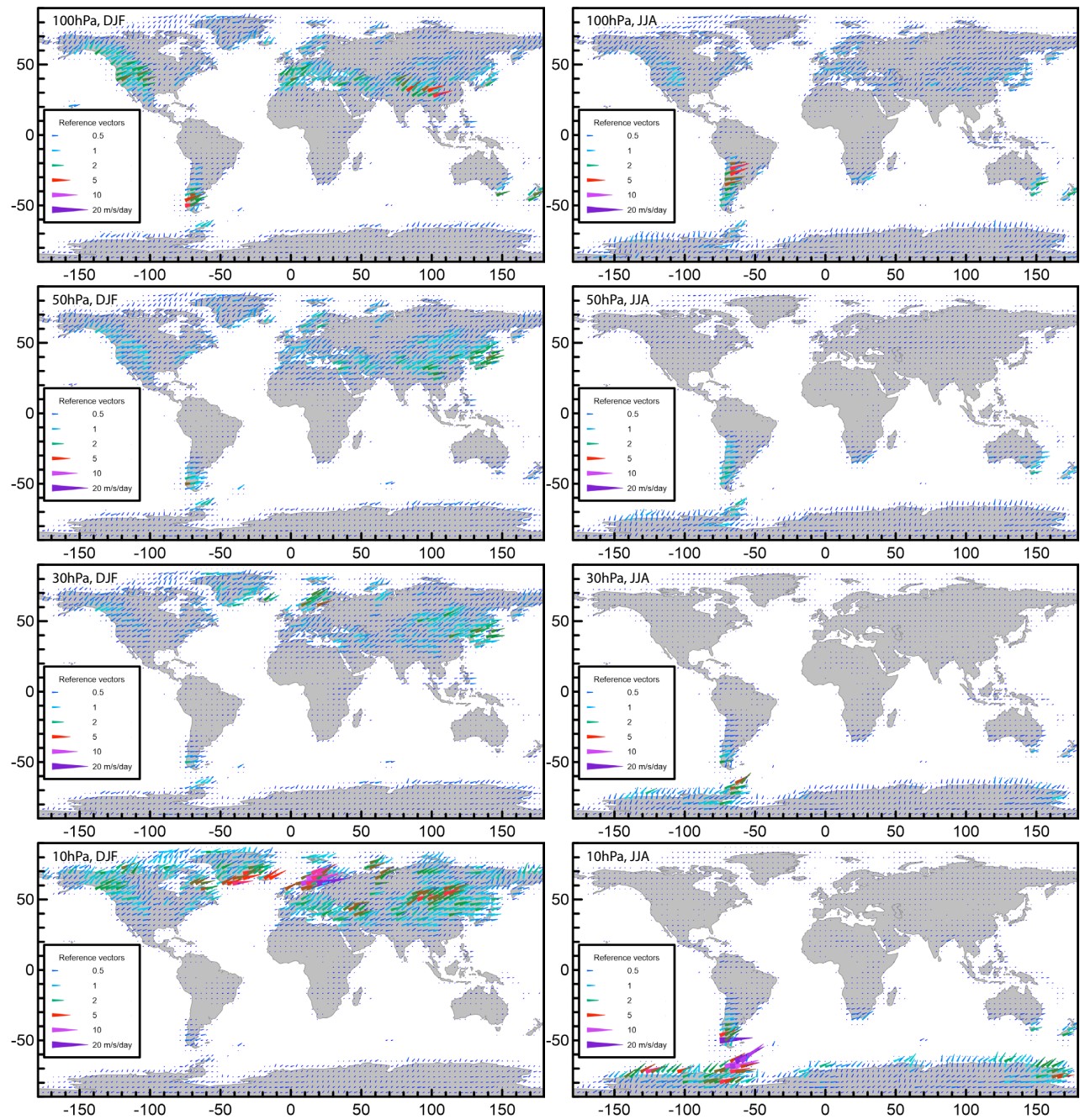

**Figure 2.** Standard deviation of the monthly series of wind tendency due to OGW [m/s/day], displayed in a vector-like form assigning the value for the eastward component to the x axis and value for the northward component to the y axis.

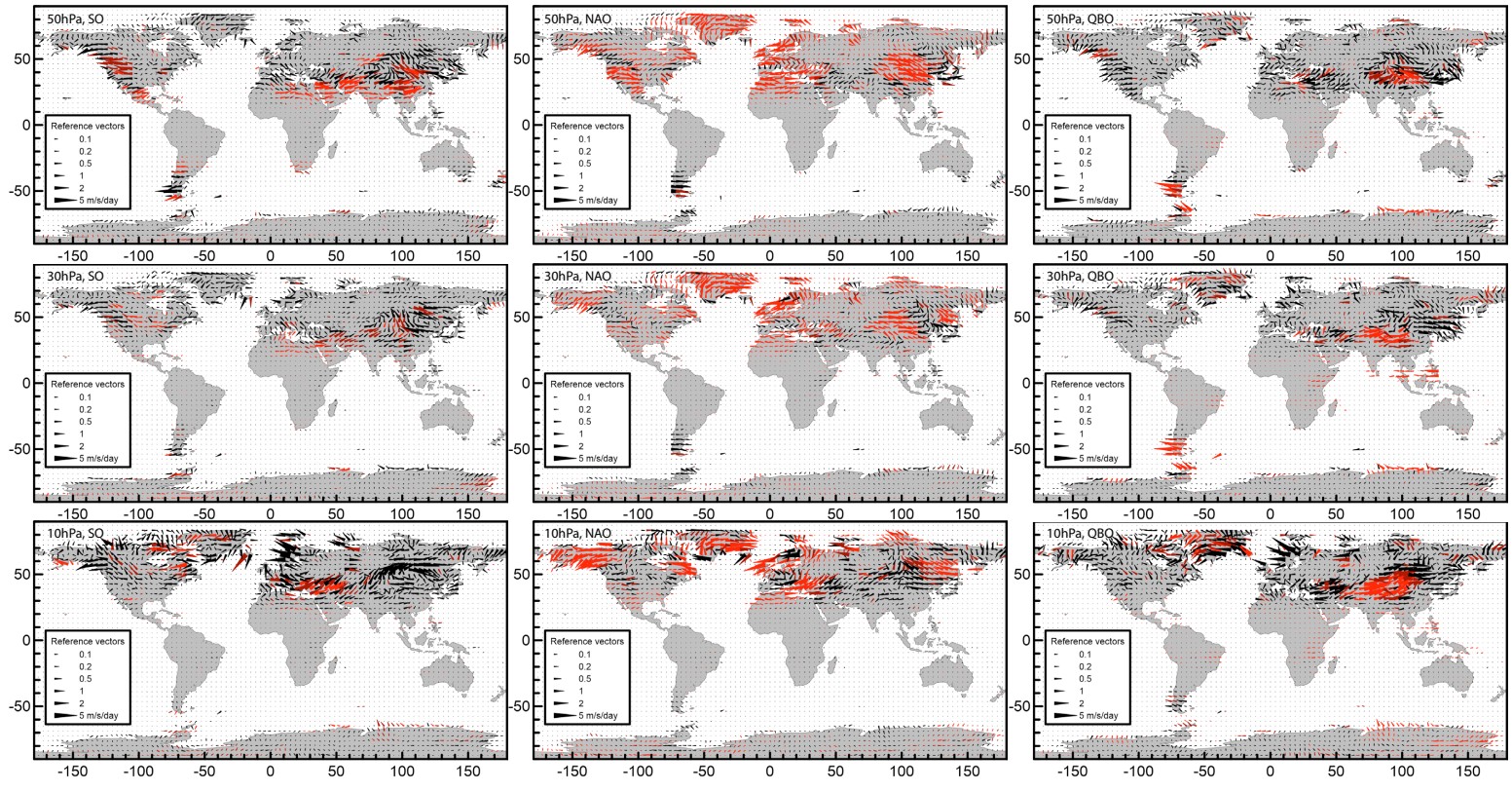

**Figure 3.** Response of the OGWD [m/s/day] at the 50 hPa (top), 30 hPa (middle) and 10 hPa (bottom) level related to the activity of the Southern Oscillation (left), North Atlantic Oscillation (center) and Quasi-Biennial Oscillation (right). The responses correspond to the increase of the oscillation index by 4x its standard deviation, i.e. to transition of the respective oscillation from highly negative to highly positive phase; red symbols pertain to locations with at least one wind tendency component response statistically significant at the 95% confidence level; bright red indicates at least one component significant at the 99% confidence level. Analysis period: 1979-2010, monthly data, DJF season.

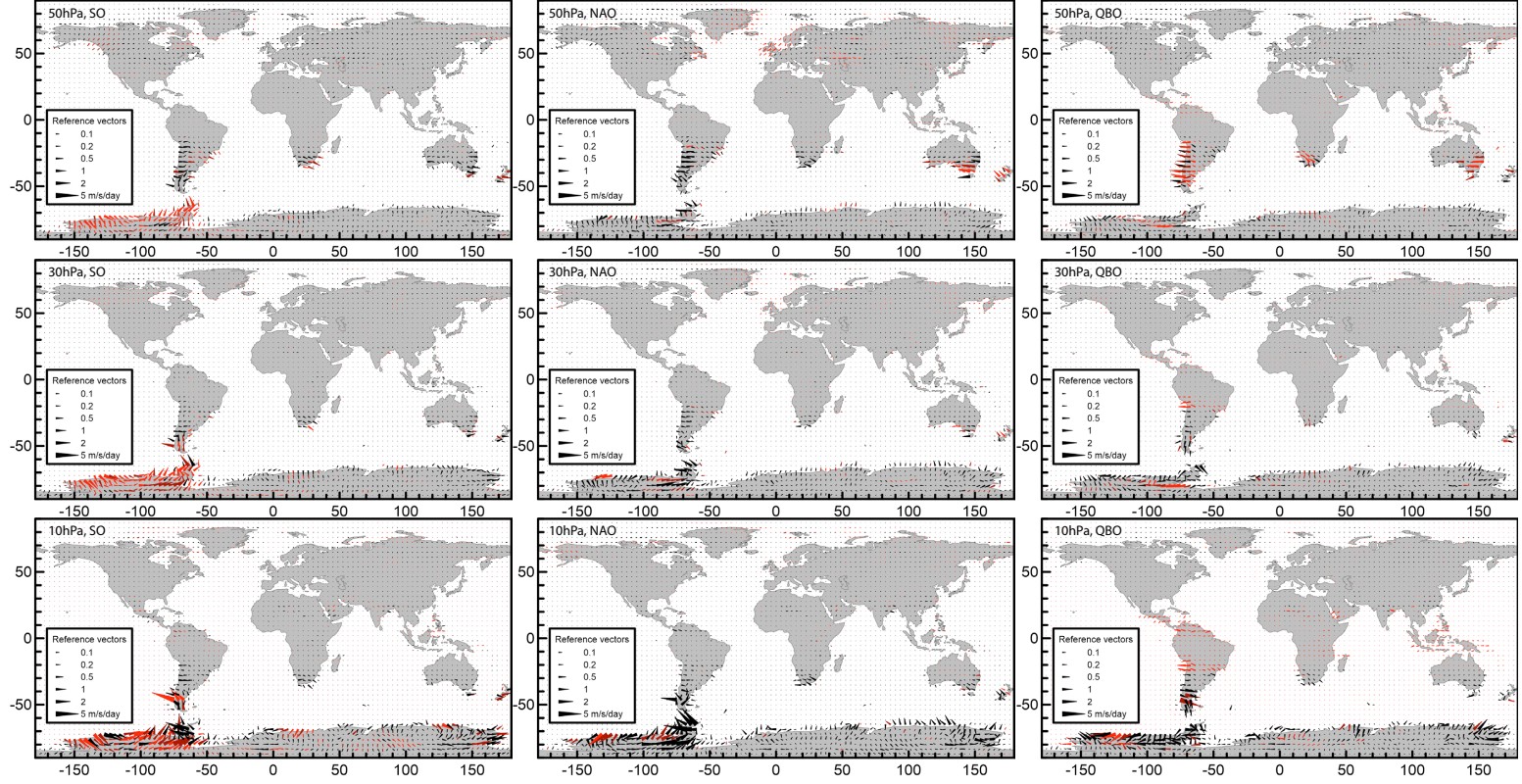

**Figure 4.** Response of the OGWD [m/s/day] at the 50 hPa (top), 30 hPa (middle) and 10 hPa (bottom) level related to the activity of the Southern Oscillation (left), North Atlantic Oscillation (center) and Quasi-Biennial Oscillation (right). The responses correspond to the increase of the oscillation index by 4x its standard deviation, i.e. to transition of the respective oscillation from highly negative to highly positive phase; red symbols pertain to locations with at least one wind tendency component response statistically significant at the 95% confidence level; bright red indicates at least one component significant at the 99% confidence level. Analysis period: 1979-2010, monthly data, JJA season

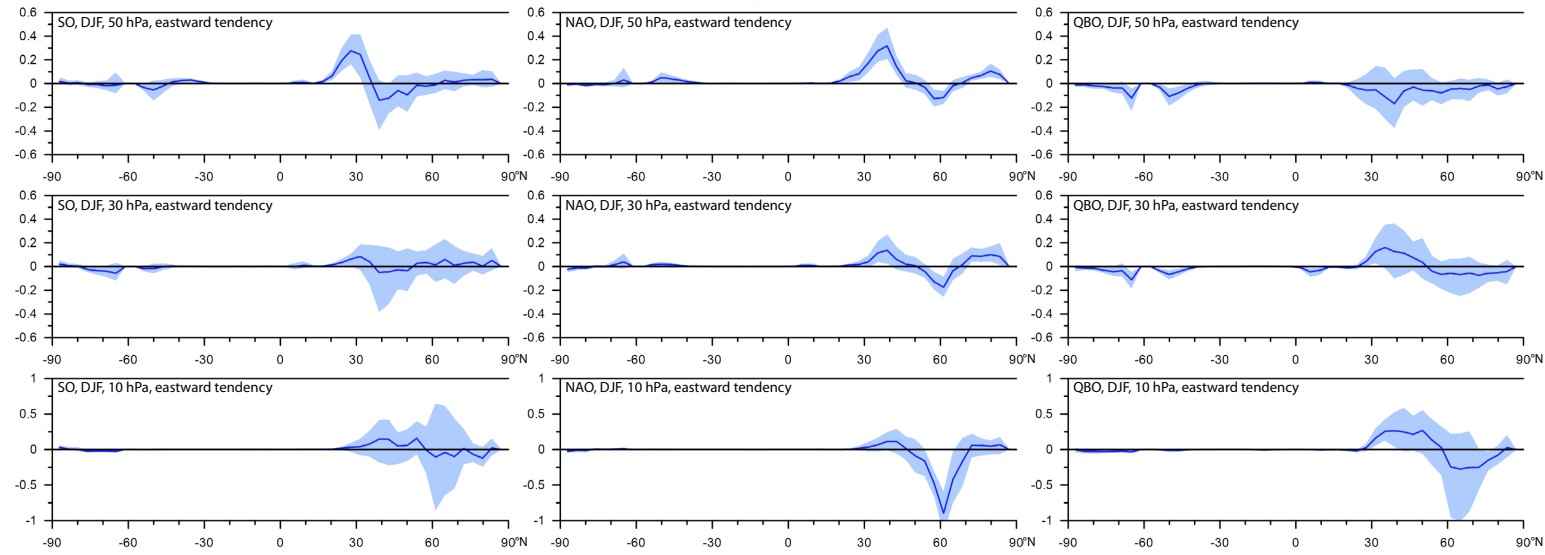

**Figure 5.** Response of the zonal mean OGWD [m/s/day] at the 50 hPa (top), 30 hPa (middle) and 10 hPa (bottom) level related to the activity of the Southern Oscillation (left), North Atlantic Oscillation (center) and Quasi-Biennial Oscillation (right). The responses correspond to the increase of the oscillation index by 4x its standard deviation, i.e. to transition of the respective oscillation from highly negative to highly positive phase; blue curve shows the signal value and blue shading illustrates the 95% confidence interval. Analysis period: 1979-2010, monthly data, DJF season.

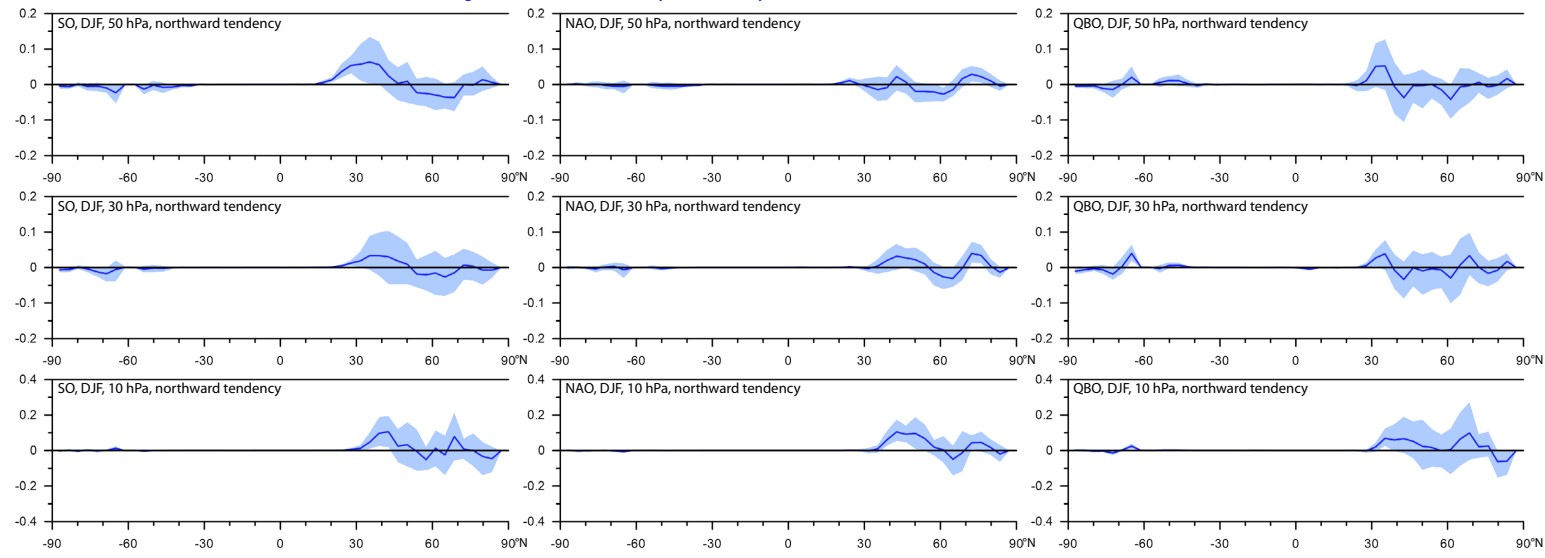

**Figure 6.** Response of the meridional mean OGWD [m/s/day] at the 50 hPa (top), 30 hPa (middle) and 10 hPa (bottom) level related to the activity of the Southern Oscillation (left), North Atlantic Oscillation (center) and Quasi-Biennial Oscillation (right). The responses correspond to the increase of the oscillation index by 4x its standard deviation, i.e. to transition of the respective oscillation from highly negative to highly positive phase; blue curve shows the signal value and blue shading illustrates the 95% confidence interval. Analysis period: 1979-2010, monthly data, DJF season.

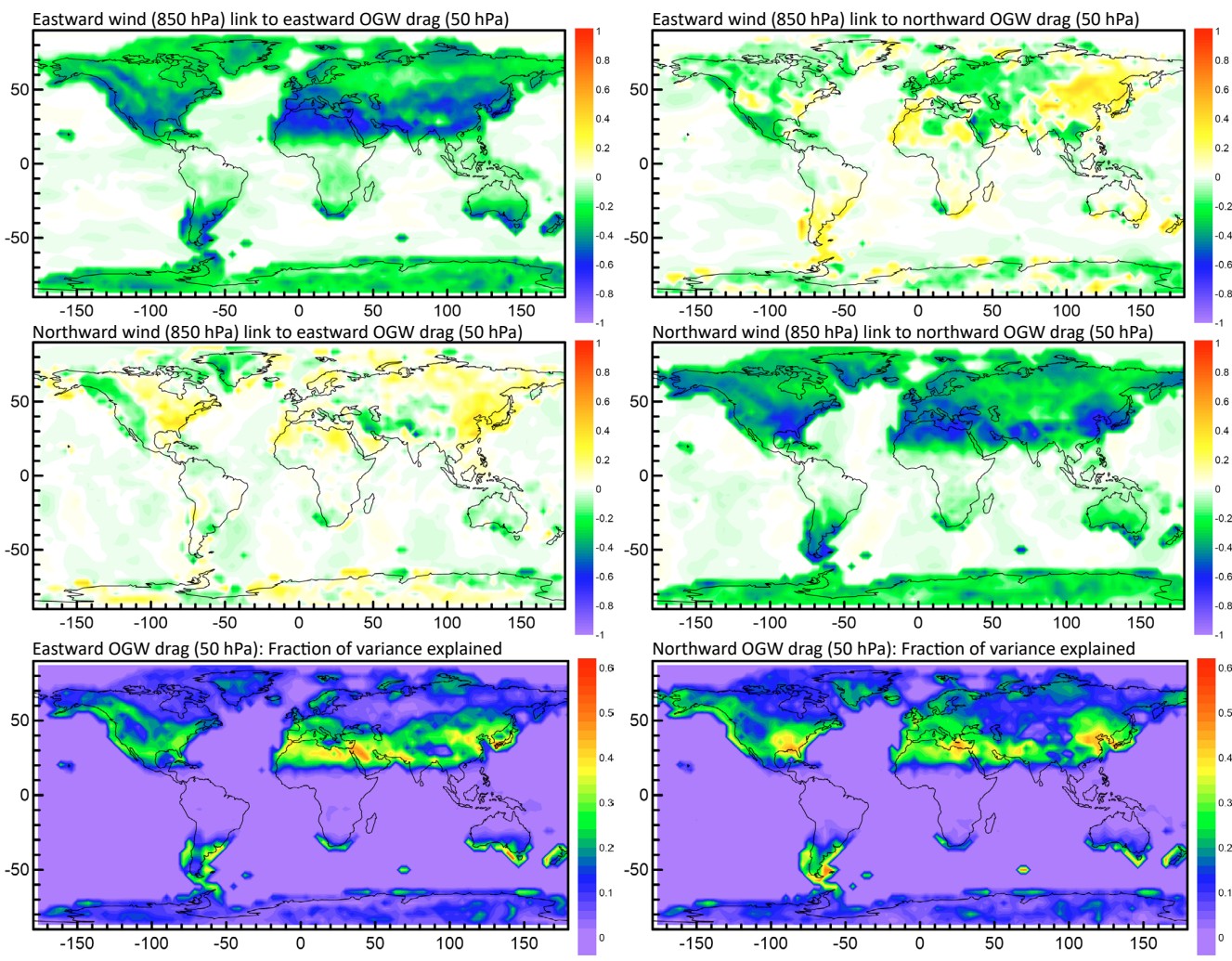

**Figure 7.** Top and middle row: Standardized regression coefficients between orographic gravity wave drag at the 50 hPa level (predictand) and eastward and northward wind components at the 850 hPa level (predictors), during the DJF season. Bottom row: coefficient of determination, i.e. fraction of total variance of OGWD explained through the regression mappings by both components of wind at the 850 hPa level. Analysis period: 1979-2010, daily data, DJF season.

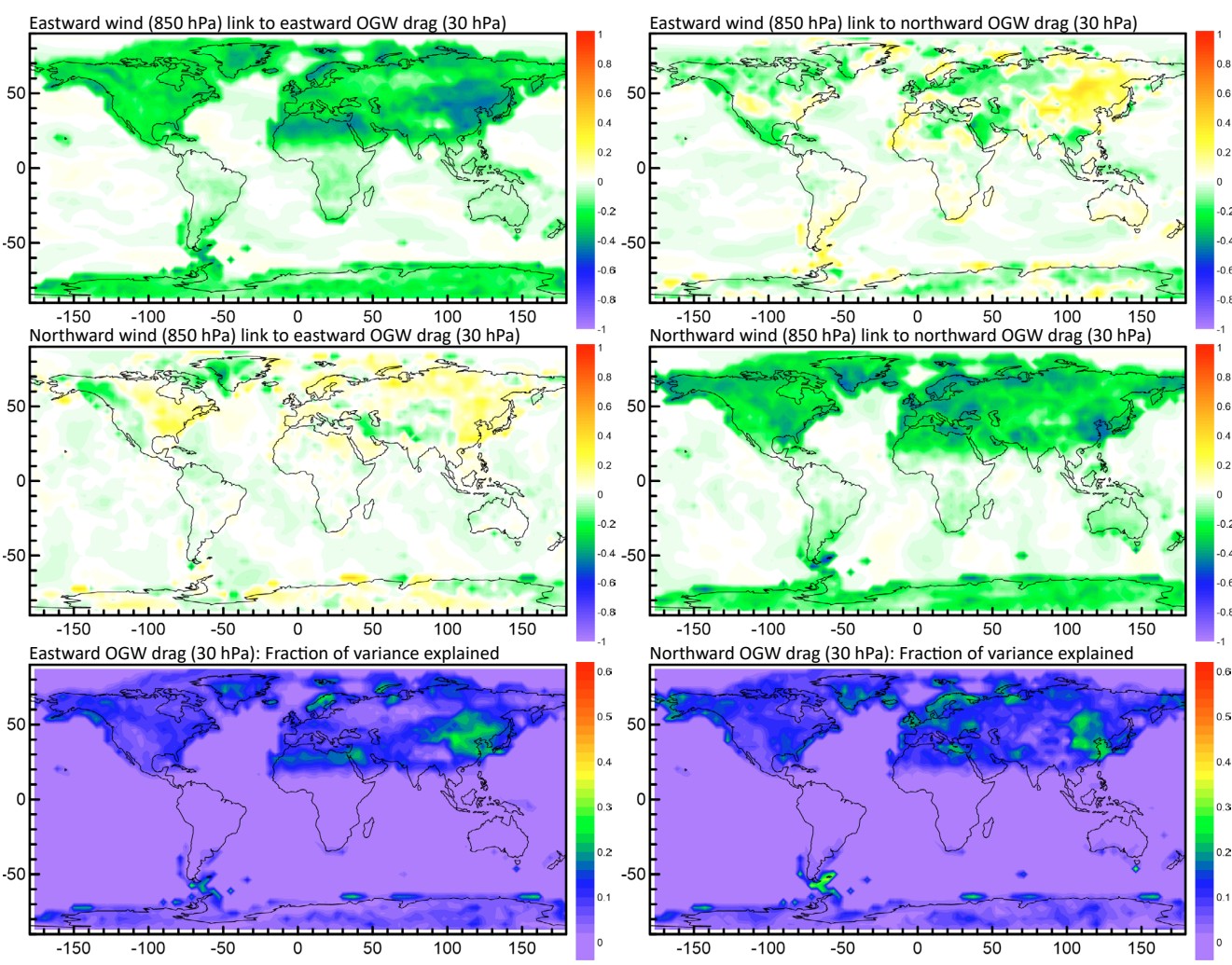

**Figure 8.** Top and middle row: Standardized regression coefficients between orographic gravity wave drag at the 30 hPa level (predictand) and eastward and northward wind components at the 850 hPa level (predictors), during the DJF season. Bottom row: coefficient of determination, i.e. fraction of total variance of OGWD explained through the regression mappings by both components of wind at the 850 hPa level. Analysis period: 1979-2010, daily data, DJF season.

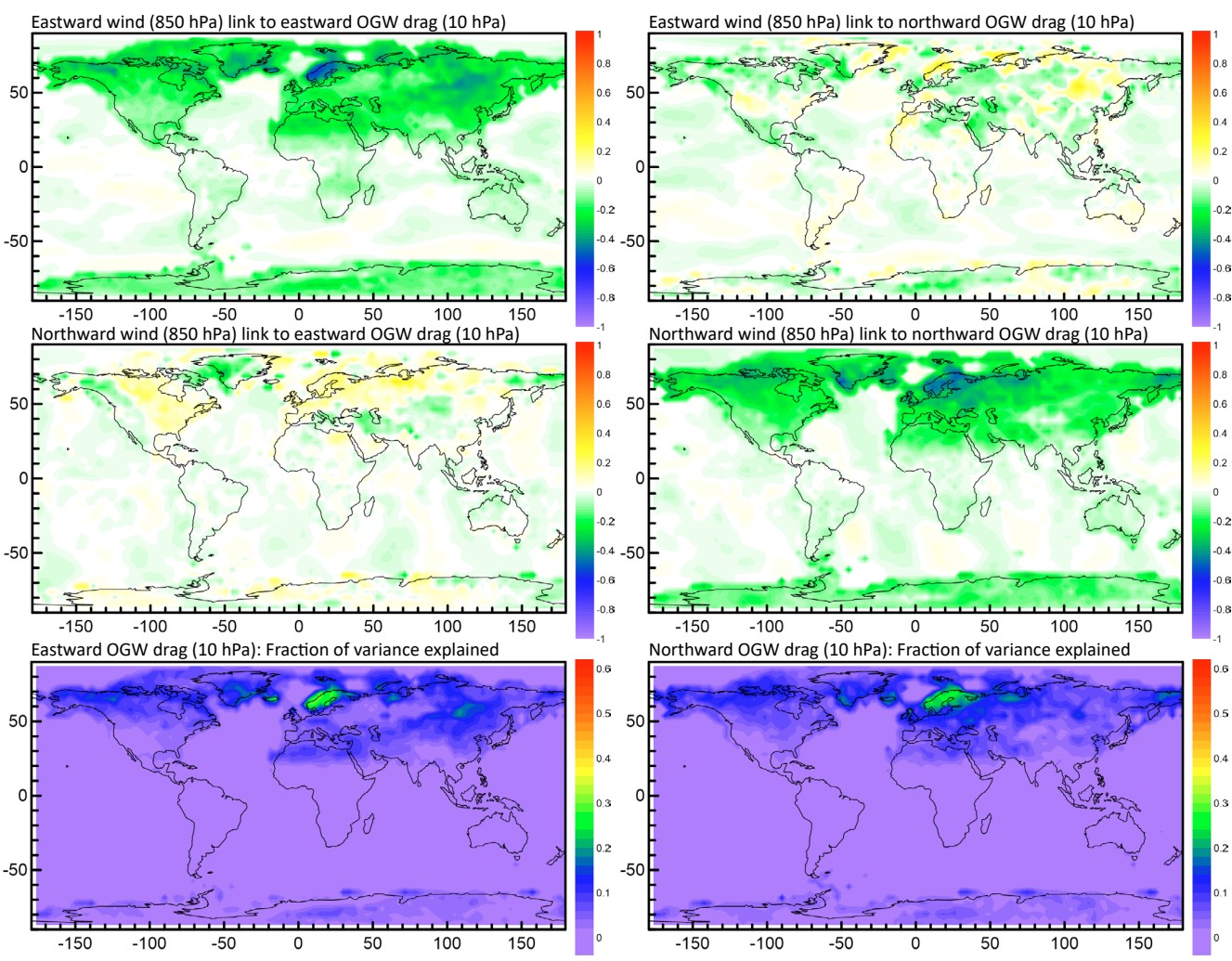

**Figure 9.** Top and middle row: Standardized regression coefficients between orographic gravity wave drag at the 10 hPa level (predictand) and eastward and northward wind components at the 850 hPa level (predictors), during the DJF season. Bottom row: coefficient of determination, i.e. fraction of total variance of OGWD explained through the regression mappings by both components of wind at the 850 hPa level. Analysis period: 1979-2010, daily data, DJF season.

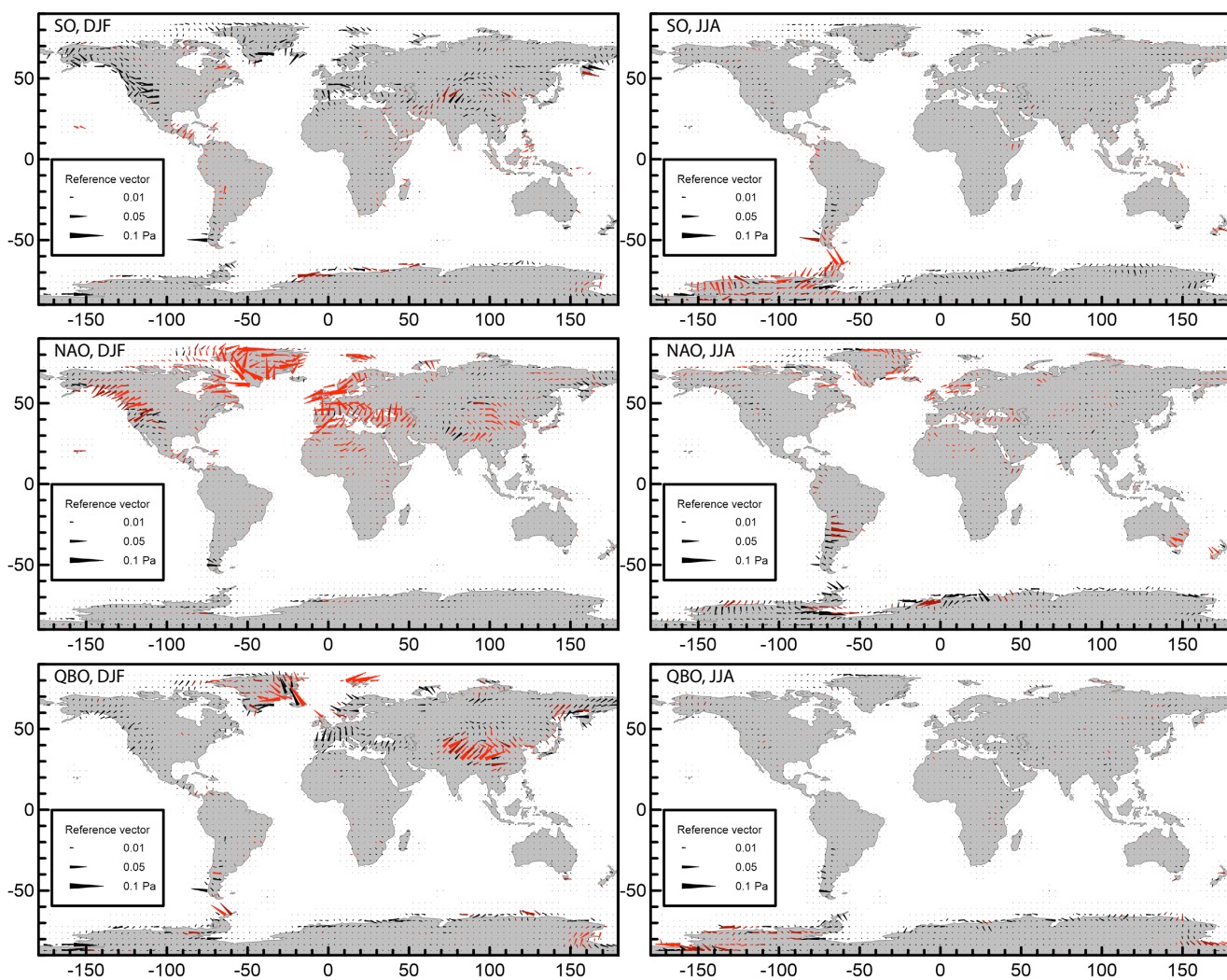

**Figure 10.** Response of the orographic GW momentum fluxes [Pa] at the 850 hPa level related to the activity of the Southern Oscillation (top), North Atlantic Oscillation (middle) and Quasi-Biennial Oscillation (bottom) during DJF (left) and JJA (right) seasons. The responses correspond to the increase of the oscillation index by 4x its standard deviation, i.e. to transition of the respective oscillation from highly negative to highly positive phase; red symbols pertain to locations with at least one orographic GW flux component response statistically significant at the 95% confidence interval. Analysis period: 1979-2010, monthly data