# Peer review of "Interannual variability of the gravity wave drag - vertical coupling and possible climate links"

_Earth System Dynamics, 2018_

## Referee Comment (RC1) · Anonymous Referee #1 · 29 Jan 2018

General Comments

In this new study, Sacha et al. discuss variability in orographic gravity wave drag (OGWD) based on a 30-year Canadian Middle Atmosphere Model (CMAM) simulation. The CMAM-sd simulation used here has specified dynamics, by nudging it to the ERA-Interim reanalysis. The authors assess the correlations between the OGWD and climate indices such as the North Atlantic Oscillation (NAO), the Quasi Biennal Oscillation (QBO), and the El Nino Southern Oscillation (ENSO) based on multiple linear regression.

Overall, the study appears to be carefully conducted and the results seem to be sound and robust. The paper is mostly well written and concise.

[Figure]

Before the paper can be accepted for publication in ESD, I would have two general comments, which perhaps could be addressed by expanding the introduction and discussion sections of the paper a bit:

1) The study is focusing on orographic gravity wave drag, which is directly provided by the OGWD parametrization of the CMAM model. However, non-orographic sources such as convection or jet and storm sources are another important source of gravity wave drag. It is pointed out that the OGWD parametrization of the CMAM-sd simulation was "tuned" to obtain more realistic circulation patterns. Does this "tuning" overemphasize the role of orographic gravity waves compared with non-orographic sources? If non-orographic sources are neglected (as I understand), how does this affect the analysis presented in this paper?

2) It would be good if this work could be put better into the context of related work. There is a number of studies discussing global climatologies of gravity wave activity in the stratosphere from observations and models, e.g.:

Gong, J., Wu, D. L., and Eckermann, S. D.: Gravity wave variances and propagation derived from AIRS radiances, Atmos. Chem. Phys., 12, 1701-1720, https://doi.org/10.5194/acp-12-1701-2012, 2012.

Geller, M.A., M.J. Alexander, P.T. Love, J. Bacmeister, M. Ern, A. Hertzog, E. Manzini, P. Preusse, K. Sato, A.A. Scaife, and T. Zhou, 2013: A Comparison between Gravity Wave Momentum Fluxes in Observations and Climate Models. J. Climate, 26, 6383–6405, https://doi.org/10.1175/JCLI-D-12-00545.1

Hoffmann, L., X. Xue, and M. J. Alexander (2013), A global view of stratospheric gravity wave hotspots located with Atmospheric Infrared Sounder observations, J. Geophys. Res. Atmos., 118, 416–434, doi:10.1029/2012JD018658.

Geller et al. (2013) showed that there are notable differences between momentum flux estimates from different models and observations. It might be good to provide more

evidence that the results from the CMAM-sd simulation are realistic. Perhaps it might be helpful to also show gravity wave momentum flux distributions from the simulation, as this can be more easily compared to other studies.

Specific Comments

- p1, l7: Be more specific about what is meant by "lower tropospheric behaviour"?

- p1, l8-9: What is meant by "have a modified impact"? Do you mean "have a modifying impact on" or simply "have impact on"?

- p2, l2-5: This first sentence is quite long. The references to Plougonven and Zhang (2014) and Alexander et al. (2009) look a bit specific considering the broad statements made here.

- p3, l5-7: How strong was the nudging? Does the CMAM-sd simulation closely follow the ERA-Interim winds and temperatures? Are the results of this study sensitive to the specific details/parameters of the nudging procedure?

- p3, l21-27: It might be worthwhile to briefly repeat/recap the definitions of the different indices?

- p5, l6-7: Is this exception of the Antarctic Peninsula due to its SW-NE orientation?

- p7, l8-11: The degree of correlation seems to decrease with height? Is this due to the stratospheric background affecting the propagation of the waves?

- p7, l22-25: Reading this, I was wondering how well the CMAM-sd simulation itself captures the different climatological patterns (NAO, SO, QBO)?

- p8, l31-32: This also triggers the question of how well the CMAM-sd simulation reflects reality?

- Fig. 1: Recent studies showed that there might be notable gravity wave activity over remote islands in the Southern Ocean, e.g., South Georgia or Kerguelen Islands. Is

the CMAM-sd simulation capable of capturing this?

Alexander, M. J., and A. W. Grimsdell (2013), Seasonal cycle of orographic gravity wave occurrence above small islands in the Southern Hemisphere: Implications for effects on the general circulation, J. Geophys. Res. Atmos., 118, 11,589–11,599, doi:10.1002/2013JD020526.

Hoffmann, L., Grimsdell, A. W., and Alexander, M. J.: Stratospheric gravity waves at Southern Hemisphere orographic hotspots: 2003–2014 AIRS/Aqua observations, Atmos. Chem. Phys., 16, 9381-9397, https://doi.org/10.5194/acp-16-9381-2016, 2016.

Technical Corrections

- The paper should be revised to fix English language issues.

- Check that acronyms are properly introduced at first occurrence (e.g. IGW or SSW).
* * *

---

## Referee Comment (RC2) · Anonymous Referee #2 · 2 Feb 2018

In this research, the authors discuss the dependence of orographic gravity wave drag (OGWD) inter-annual variability on the tropospheric climate modes through the 30 year simulation using Canadian Middle Atmospheric Model (CMAM), and assess the potential relationship between OGWD and large-scale climate modes such as North Atlantic Oscillation (NAO), the Quasi-Biennal Oscillation (QBO), and the El Nino Southern Oscillation (ENSO) using multiple linear regression. It was argued that the orographic gravity wave and gravity waves in general can be a quick mediator of tropospheric variability into the stratosphere. The topic is very interesting, however, the following comments should be addressed before publication in ESD.

Major comments:

1. The author solely relies on the orographic gravity wave drag parametrization scheme

of the CMAM model to examine the inter-annual variability of the orographic gravity wave drag. However, the tuning procedure as mentioned in the research may potentially overestimate the role of orographic gravity wave as compared to non-orographic source such as convections. How does the choice of tuning parameters in the orographic gravity wave drag scheme affect the conclusions in this research?

2. The QBO in this simulation is potentially affected by nudging. How is QBO represented in the simulation as compared to the observation? And how sensitive is the relationship between orographic gravity wave drag and QBO to nudging?

Minor comments:

1. The wind vectors in figure 1, 4 and 5 are too thin to see, it help vision if the wind vectors are drawn thicker.

2. In figure 2, it would be more concise if the standard deviation of the wind vector amplitude (norm of the wind vector) due to orographic gravity drag rather than both zonal and meridional components are shown. This is also the case for Figure 3.

3. In Figure 8, 9 and 10, the author mentioned the fractions are explained by "both component", does this mean the combined norm variance of the 850-hPa wind vector?

4. Acronyms such as PW, SSW, IGW needs clarification.

---

## Author Comment (AC1) · 21 Feb 2018

**Responses to the referee's comment on paper *" Interannual variability of the gravity wave drag - vertical coupling and possible climate links"* by Petr Sacha, Jiri Miksovsky, and Petr Pisoft.**

We would like to thank the referee for taking the time to review our manuscript. We greatly appreciate the insightful and constructive comments, which we address in our responses below.

[Figure]

**Major comments:**

1) *The author solely relies on the orographic gravity wave drag parameterization scheme of the CMAM model to examine the inter-annual variability of the orographic gravity wave drag. However, the tuning procedure as mentioned in the research may potentially overestimate the role of orographic gravity wave as compared to non-orographic source such as convections. How does the choice of tuning parameters in the orographic gravity wave drag scheme affect the conclusions in this research?*

Thank you for this comment; it is a very good point. The settings of the OGWD and NOGWD parameterization scheme used in the CMAM-sd simulation are discussed in details in McLandress et al. (2013) and we will add a paragraph in the revised manuscript to make a clear summary discussing limitations connected to our results.

The tuning of OGWD consists of arbitrary choosing a value of dimensionless parameters controlling the total value of launch momentum and the vertical flux of horizontal momentum. As an indirect effect the breaking level of the waves is influenced by this setting. The nudging procedure (its influence is discussed in more detail in reply to REF1) helps to reach realistic distributions of momentum fluxes but the breaking levels and OGWD value are largely influenced by this arbitrary tuning.

As for the overestimation of the OGW relative to NOGW role, we entirely agree and it is certain that at the analyzed levels the overestimation is present. We will stress this out more in the revised version together with statement that our conclusions are directly applicable at the model atmosphere only and thus having indirect implications for the real atmosphere (as already stated in the current version of the discussion).

Let us follow with an excerpt from the reply to REF1: It would be very interesting to look at NOGWD variations connected with variability of jets, fronts

etc. However, the CMAM NOGWD scheme (Scinocca, 2003) is based on launching a globally uniform isotropic NOGW spectrum in four cardinal horizontal directions at approximately 125 hPa. The aim is to produce reasonable seasonal evolution of the zonal mean zonal temperature and winds in the mesosphere and the zonal and meridional asymmetry stems from propagation effects only. Regarding NOGWD, we have produced the same analysis as for the OGWD but due to the above mentioned reasons the resulting fields are highly zonally symmetrical and weaker in magnitude compared to the OGWD and so we decided not to show them in the manuscript. However, we attach selected figures as a supplement to this response.

We expect the NOGWD interannual variability in the upper troposphere-lower stratosphere region to show highly zonally asymmetric behavior (storm track shifts, distribution of convection, etc.) and in our future research we would like to analyze a dataset that would at least roughly capture this.

2) *The QBO in this simulation is potentially affected by nudging. How is QBO represented in the simulation as compared to the observation? And how sensitive is the relationship between orographic gravity wave drag and QBO to nudging?*

The QBO representation is close to reality due to the nudging (for instance, the Pearson correlation of the CMAM-based and observational QBO index is 0.96 over the 1979-2010 period) and therefore also the modulation of the OGWD by QBO should be realistic. OGWD is very sensitive to the QBO mainly due to modulation of the background for the GW propagation. In our results we can also see some QBO influence on OGWD sourcing - probably due to polar vortex teleconnection. It would be highly interesting to look at the QBO influence on NOGW sourcing (convection etc.), but, as discussed above, we are not able to assess this in CMAM.

**Minor comments:**

1) *The wind vectors in figure 1, 4 and 5 are too thin to see, it help vision if the wind vectors are drawn thicker.*

    The figures will be adjusted in the revised version of the paper.

2) *In figure 2, it would be more concise if the standard deviation of the wind vector amplitude (norm of the wind vector) due to orographic gravity drag rather than both zonal and meridional components are shown. This is also the case for Figure 3.*

    The figures will be adjusted in the revised version of the paper.

3) *In Figure 8, 9 and 10, the author mentioned the fractions are explained by "both component", does this mean the combined norm variance of the 850-hPa wind vector?*

    The value in question represents a coefficient of determination ($R^2$) associated with the multiple regression mapping using both components of 850 hPa wind (eastward and northward) as predictors; the captions of the figures have been updated to better explain this.

4) *Acronyms such as PW, SSW, IGW needs clarification.*

    The acronyms will be correctly introduced in the revised version of the paper.

[Figure]

**Fig. 1.** Response of the non-orographic GWD [m/s/s] at the 50 hPa level related to the activity of the Southern Oscillation (left), North Atlantic Oscillation (right) and Quasi-Biennial Oscillation (bottom)

---

## Author Comment (AC2) · 21 Feb 2018

**Responses to the referee's comment on paper** *" Interannual variability of the gravity wave drag - vertical coupling and possible climate links"* **by Petr Sacha, Jiri Miksovsky, and Petr Pisoft.**

We would like to thank the referee for taking the time to review our manuscript. We greatly appreciate the insightful and constructive comments, which we address in our responses below.

[Figure]

**General comments:**

1) *The study is focusing on orographic gravity wave drag, which is directly provided by the OGWD parameterization of the CMAM model. However, non-orographic sources such as convection or jet and storm sources are another important source of gravity wave drag. It is pointed out that the OGWD parameterization of the CMAM-sd simulation was "tuned" to obtain more realistic circulation patterns. Does this "tuning" overemphasize the role of orographic gravity waves compared with non-orographic sources? If non-orographic sources are neglected (as I understand), how does this affect the analysis presented in this paper?*

Thank you for this comment; it is a very good point. The settings of the OGWD and NOGWD parameterization scheme used in the CMAM-sd simulation are discussed in details in McLandress et al. (2013) and we will add a paragraph in the revised manuscript to make a clear summary discussing limitations connected to our results.

The dynamical role of the OGWD relative to the NOGWD at the vertical domain of our analysis is most likely overestimated compared to the current consensus on the GW impacts on the stratosphere. We will stress this out more in the revised version together with statement that our conclusions are directly applicable at the model atmosphere only and thus having indirect implications for the real atmosphere (as already stated in the current version of the discussion).

It would be very interesting to look at NOGWD variations connected with variability of jets, fronts etc. However, the CMAM NOGWD scheme (Scinocca, 2003) is based on launching a globally uniform isotropic NOGW spectrum in four cardinal horizontal directions at approximately 125 hPa. The aim is to produce reasonable seasonal evolution of the zonal mean zonal temperature and winds in the mesosphere and the zonal and meridional asymmetry stems from propagation effects only. Regarding NOGWD, we have produced the same analysis as for the OGWD but due to the above mentioned reasons the resulting fields are highly zonally symmetrical and weaker in magnitude compared to the OGWD and so we decided not to show them in the manuscript. However, we attach selected figures as a supplement to this response (see Fig. 1 below).

2) *It would be good if this work could be put better into the context of related work. There is a number of studies discussing global climatologies of gravity wave activity in the stratosphere from observations and models, e.g. ... Geller et al. (2013) showed that there are notable differences between momentum flux estimates from different models and observations. It might be good to provide more evidence that the results from the CMAM-sd simulation are realistic. Perhaps it might be helpful to also show gravity wave momentum flux distributions from the simulation, as this can be more easily compared to other studies.*

We thank the referee for pointing out additional studies that can improve our assessment of realisticity of the CMAM-sd OGWD distribution and also we thank for the idea of comparing the momentum fluxes. We will adopt that and the results will be discussed in the revised version of the paper.

**Specific comments:**

1) *Be more specific about what is meant by "lower tropospheric behavior"?*

The OGWD variability is shown to be induced by lower tropospheric wind variations in a large part. There was also significant variability detected in near surface OGW momentum fluxes.

2) *What is meant by "have a modified impact"? Do you mean "have a modifying impact on" or simply "have impact on"?*

We meant that there were modifications of the GWs impact. The statement will be adjusted in the following way.

We argue that the orographic gravity waves (OGWs) and GWs in general can be a quick mediator of the tropospheric variability into the stratosphere as the modifications of the OGWD distribution can result in different impacts on the stratospheric dynamics during different phases of the studied climate oscillations.

3) *This first sentence is quite long. The references to Plougonven and Zhang (2014) and Alexander et al. (2009) look a bit specific considering the broad statements made here.*

The statement will be adjusted in the following way.

Although the gravity wave (GW) sourcing (e.g. adjustment processes, Plougonven and Zhang, 2014), propagation and breaking is governed to some extent by processes in the stratosphere, there is a significant portion of the IGW spectra created in the troposphere (mostly orography and convection, Alexander et al., 2009). The highest amplitude upward propagating modes can break already in the troposphere and lower or middle stratosphere (Fritts et al., 2016).

- Fritts, D.C., R.B. Smith, M.J. Taylor, J.D. Doyle, S.D. Eckermann, A. Dörnbrack, M. Rapp, B.P. Williams, P. Pautet, K. Bossert, N.R. Criddle, C.A. Reynolds, P.A. Reinecke, M. Uddstrom, M.J. Revell, R. Turner, B. Kaifler, J.S. Wagner, T. Mixa, C.G. Kruse, A.D. Nugent, C.D. Watson, S. Gisinger, S.M. Smith, R.S. Lieberman, B. Laughman, J.J.

Moore, W.O. Brown, J.A. Haggerty, A. Rockwell, G.J. Stossmeister, S.F. Williams, G. Hernandez, D.J. Murphy, A.R. Klekociuk, I.M. Reid, and J. Ma, 2016: The Deep Propagating Gravity Wave Experiment (DEEPWAVE): An Airborne and Ground-Based Exploration of Gravity Wave Propagation and Effects from Their Sources throughout the Lower and Middle Atmosphere. Bull. Amer. Meteor. Soc., 97, 425–453, https://doi.org/10.1175/BAMS-D-14-00269.1

4) *How strong was the nudging? Does the CMAM-sd simulation closely follow the ERA-Interim winds and temperatures? Are the results of this study sensitive to the specific details/parameters of the nudging procedure?*

The nudging procedure is examined in detail in McLandress et al. (2014) and described in McLandress et al. (2013). Below 1 hPa, in spectral space (for horizontal scales with wavenumber lower than 21), CMAM is nudged to the 6 hourly horizontal winds and temperatures from ERA Interim. This would definitely be a problem, if we would analyze the GWD impact on the circulation. This impact is weakened by the relaxation. But, as we have only analyzed the OGWD interannual variability, the nudging procedure is very advantageous for us (compared to a free running model). Since the winds and temperatures in the lower atmosphere have a strong influence on the propagation and absorption of gravity waves the distribution of parameterized (and resolved) gravity wave fluxes in CMAM-sd can match the real fluxes (McLandress et al., 2013). This is true, of course, having the tuning and other specifics of GWD parameterization in mind.

5) *It might be worthwhile to briefly repeat/recap the definitions of the different indices?*

The definitions together with description of the utilized data will be added in the revised version of the paper.

[Figure]

6) *Is this exception of the Antarctic Peninsula due to its SW-NE orientation?*

We would not say an exception in general (only the strongest) but we agree with the referee that the topography orientation (relative to surface winds) is important. The orientation of the momentum flux in the OGWD parametrization in CMAM is a function of the near-surface wind direction relative to the orientation of the topography (McLandress et al., 2013). In Sacha et al. (2016) we highlighted the fact that the dynamical effects of the meridional GWD component do not receive a sufficient attention and are quite uncertain to date.

- Sacha, P., Lilienthal, F., Jacobi, C., and Pisoft, P.: Influence of the spatial distribution of gravity wave activity on the middle atmospheric dynamics, Atmos. Chem. Phys., 16, 15755-15775, https://doi.org/10.5194/acp-16-15755-2016, 2016.

7) *The degree of correlation seems to decrease with height? Is this due to the stratospheric background affecting the propagation of the waves?*

Yes, this is due to the background. It is well manifested in the eastern Asia region where this hotspot is stronger at 50 hPa and when the stratospheric conditions are not advantageous for breaking at 50 hPa there is more breaking at 30 hPa. This results in the OGWD at 30hPa in this region having less correlation with tropospheric conditions because the stratospheric variability between 50 and 30 hPa is important.

But this is not the case for the Scandinavian hotspot that is climatologically strongest at 10 hPa and the fraction explained by near surface winds is lower at 50 and 30 hPa. At those levels the waves dissipate less frequently - predominantly in dependence on the stratospheric variability.

Based on the referee comment, we will add following statement to the manuscript:

p7l9 This is due to the stratospheric background affecting the critical line occurrence and propagation of the GWs between 50 and 30 hPa in the eastern Asia region.

8) *Reading this, I was wondering how well the CMAM-sd simulation itself captures the different climatological patterns (NAO, SO, QBO)?*

While there are some differences between the time series of CMAM-based and observational counterparts, their similarity is generally strong, and the Pearson correlation (i.e., a correlation measure best suited for quantifying the links explored by linear regression analysis) between them is high. See Fig. 2 below for an example of temporal variability of the CMAM-simulated and observational indices of the Southern Oscillation.

9) *This also triggers the question of how well the CMAM-sd simulation reflects reality?*

As with all model-based frameworks, the issue of reliable reproduction of the observed climate is quite complex and many statistics can be considered for validation. Our tests concentrated primarily on the ability of CMAM to realistically reproduce the spatial patterns of response of lower tropospheric characteristics to the phases of the oscillation considered in our analysis (NAO, SO/ENSO, QBO); the results suggested a high degree of match between the CMAM-based and observational data (see Fig. 3 below for a sample of the results, summarizing the 850 hPa temperature response to SO and NAO).

[Figure]

10) *Recent studies showed that there might be notable gravity wave activity over re-mote islands in the Southern Ocean, e.g., South Georgia or Kerguelen Islands. Is the CMAM-sd simulation capable of capturing this?*

In the Figure 1, 50 hPa, DJF (!) we can see that there is some moderately strong significant OGWD over those remote islands. However, we must note that those are monthly mean values only, not reflecting the intermittent nature of the GWs. Our study is focused globally, but the fact that CMAM is able to get some pronounced OGWD above such small islands is good news for possible future research of individual hotspots.

- Alexander, M. J., and A. W. Grimsdell (2013), Seasonal cycle of oro-graphic gravity wave occurrence above small islands in the Southern Hemisphere: Implications for effects on the general circulation, J. Geo-phys. Res. Atmos., 118, 11,589–11,599, doi:10.1002/2013JD020526.
- Hoffmann, L., Grimsdell, A. W., and Alexander, M. J.: Stratospheric gravity waves at Southern Hemisphere orographic hotspots: 2003–2014 AIRS/Aqua observations, At- mos. Chem. Phys., 16, 9381-9397, https://doi.org/10.5194/acp-16-9381-2016, 2016.

[Figure]

**Fig. 1.** Response of the non-orographic GWD [m/s/s] at the 50 hPa level related to the activity of the Southern Oscillation (left), North Atlantic Oscillation (right) and Quasi-Biennial Oscillation (bottom)

[Figure]

**Fig. 2.** Southern Oscillation index calculated as a normalized sea level pressure difference between Darwin and Tahiti, derived from CMAM data (red line) and from direct observations (blue; CRU)

**Fig. 3.** 850 hPa temperature response to SO (top) and NAO (down) change from highly nega-
tive to highly positive phase, evaluated through multiple linear regression.

---

## Short Comment (SC1) · 22 Feb 2018

Regarding our answer to the specific comment 9 it is necessary to add that the plots in Fig. 3 supplemented to our answers were compared to plots that were derived in same way but using various observational datasets. For more details, please see Miksovsky, J., Holtanova, E., and Pisoft, P.: Imprints of climate forcings in global gridded temperature data, Earth Syst. Dynam., 7, 231-249, https://doi.org/10.5194/esd-7-231-2016, 2016

---

## Author Response (AR1)

**Reply to the reviewers' comments on our manuscript:**
Interannual variability of the gravity wave drag - vertical coupling and possible climate links

Petr Sacha, Jiri Miksovsky and Petr Pisoft

We thank the reviewers for the positive judgment on our manuscript and their constructive comments. We took all the reviewers' comments into account when preparing the revised version of the manuscript. Below we address the reviewers' comments point by point and enclose the manuscript version where the changes are highlighted.

**Referee #1**

**General comments:**

1) *The study is focusing on orographic gravity wave drag, which is directly provided by the OGWD parameterization of the CMAM model. However, non-orographic sources such as convection or jet and storm sources are another important source of gravity wave drag. It is pointed out that the OGWD parameterization of the CMAM-sd simulation was "tuned" to obtain more realistic circulation patterns. Does this "tuning" overemphasize the role of orographic gravity waves compared with non-orographic sources? If non-orographic sources are neglected (as I understand), how does this affect the analysis presented in this paper?*

Thank you for this comment; it is a very good point. The settings of the OGWD and NOGWD parameterization scheme used in the CMAM-sd simulation are discussed in details in McLandress et al. (2013) and we added a paragraph in the revised manuscript to make a clear summary discussing limitations connected to our results.

The dynamical role of the OGWD relative to the NOGWD at the vertical domain of our analysis is most likely overestimated compared to the current consensus on the GW impacts on the stratosphere. We stressed this out more in the revised version together with statement that our conclusions are directly applicable at the model atmosphere only and thus having indirect implications for the real atmosphere.

It would be very interesting to look at NOGWD variations connected with variability of jets, fronts etc. However, the CMAM NOGWD scheme (Scinocca, 2003) is based on launching a globally uniform isotropic NOGW spectrum in four cardinal horizontal directions at approximately 125 hPa. The aim is to produce reasonable seasonal evolution of the zonal mean zonal temperature and winds in the mesosphere and the zonal and meridional asymmetry stems from propagation effects only. Regarding NOGWD, we have produced the same analysis as for the OGWD but due to the above mentioned reasons the resulting fields are highly zonally symmetrical and weaker in magnitude compared to the OGWD and so we decided not to show them in the manuscript. However, we attach selected figure to the manuscript supplement (Fig. S5).

2) *It would be good if this work could be put better into the context of related work. There is a number of studies discussing global climatologies of gravity wave activity in the stratosphere from observations and models, e.g…. Geller et al. (2013) showed that there are notable differences between momentum flux estimates from different models and observations. It might be good to provide more evidence that the results from the CMAM-sd*

*simulation are realistic. Perhaps it might be helpful to also show gravity wave momentum flux distributions from the simulation, as this can be more easily compared to other studies.*

We thank the referee for pointing out additional studies that can improve our assessment of realisticity of the CMAM-sd OGWD distribution and also we thank for the idea of comparing the momentum fluxes. We adopt that and the results are discussed in the revised version of the paper.

**Specific comments:**

*1) Be more specific about what is meant by "lower tropospheric behavior"?*

The OGWD variability is shown to be induced by lower tropospheric wind variations in a large part. There was also significant variability detected in near surface OGW momentum fluxes.

*2) What is meant by "have a modified impact"? Do you mean "have a modifying impact on" or simply "have impact on"?*

We meant that there are modifications of the GWs impact. The statement is adjusted in the following way.

We argue that the orographic gravity waves (OGWs) and GWs in general can be a quick mediator of the tropospheric variability into the stratosphere as the modifications of the OGWD distribution can result in different impacts on the stratospheric dynamics during different phases of the studied climate oscillations.

*3) This first sentence is quite long. The references to Plougonven and Zhang (2014) and Alexander et al. (2009) look a bit specific considering the broad statements made here.*

The statement is adjusted in the following way.

Although the gravity wave (GW) sourcing (e.g. adjustment processes, Plougonven and Zhang, 2014), propagation and breaking is governed to some extent by processes in the stratosphere, there is a significant portion of the IGW spectra created in the troposphere (mostly orography and convection, Alexander et al., 2009). The highest amplitude upward propagating modes can break already in the troposphere and lower or middle stratosphere (Fritts et al., 2016).

- Fritts, D.C., R.B. Smith, M.J. Taylor, J.D. Doyle, S.D. Eckermann, A. Dörnbrack, M. Rapp, B.P. Williams, P. Pautet, K. Bossert, N.R. Criddle, C.A. Reynolds, P.A. Reinecke, M. Uddstrom, M.J. Revell, R. Turner, B. Kaifler, J.S. Wagner, T. Mixa, C.G. Kruse, A.D. Nugent, C.D. Watson, S. Gisinger, S.M. Smith, R.S. Lieberman, B. Laughman, J.J. Moore, W.O. Brown, J.A. Haggerty, A. Rockwell, G.J. Stossmeister, S.F. Williams, G. Hernandez, D.J. Murphy, A.R. Klekociuk, I.M. Reid, and J. Ma, 2016: The Deep Propagating Gravity Wave Experiment (DEEPWAVE): An Airborne and Ground-Based Exploration of Gravity Wave Propagation and Effects from Their Sources throughout the Lower and Middle Atmosphere. Bull. Amer. Meteor. Soc., 97, 425–453, https://doi.org/10.1175/BAMS-D-14-00269.1

*4) How strong was the nudging? Does the CMAM-sd simulation closely follow the ERA-Interim winds and temperatures? Are the results of this study sensitive to the specific details/parameters of the nudging procedure?*

The nudging procedure is examined in detail in McLandress et al. (2014) and described in McLandress et al. (2013). Below 1 hPa, in spectral space (for horizontal scales with wavenumber lower than 21), CMAM is nudged to the 6 hourly horizontal winds and temperatures from ERA Interim. This would definitely be a problem, if we would analyze the GWD impact on the circulation. This impact is weakened by the relaxation. But, as we have only analyzed the OGWD interannual variability, the nudging procedure is very advantageous for us (compared to a free running model). Since the winds and temperatures in the lower atmosphere have a strong influence on the propagation and absorption of gravity waves the distribution of parameterized (and resolved) gravity wave fluxes in CMAM-sd can match the real fluxes (McLandress et al., 2013). This is true, of course, having the tuning and other specifics of GWD parameterization in mind.

*5) It might be worthwhile to briefly repeat/recap the definitions of the different indices?*

The definitions together with description of the utilized data are added in the revised version of the paper and the respective section of the text in Sect. 2.2 has been expanded.

*6) Is this exception of the Antarctic Peninsula due to its SW-NE orientation?*

We would not say an exception in general (only the strongest) but we agree with the referee that the topography orientation (relative to surface winds) is important. The orientation of the momentum flux in the OGWD parametrization in CMAM is a function of the near-surface wind direction relative to the orientation of the topography (McLandress et al., 2013). In Sacha et al. (2016) we highlighted the fact that the dynamical effects of the meridional GWD component do not receive a sufficient attention and are quite uncertain to date.

- Sacha, P., Lilienthal, F., Jacobi, C., and Pisoft, P.: Influence of the spatial distribution of gravity wave activity on the middle atmospheric dynamics, Atmos. Chem. Phys., 16, 15755-15775, https://doi.org/10.5194/acp-16-15755-2016, 2016.

*7) The degree of correlation seems to decrease with height? Is this due to the stratospheric background affecting the propagation of the waves?*

Yes, this is due to the background. It is well manifested in the eastern Asia region where this hotspot is stronger at 50 hPa and when the stratospheric conditions are not advantageous for breaking at 50 hPa there is more breaking at 30 hPa. This results in the OGWD at 30hPa in this region having less correlation with tropospheric conditions because the stratospheric variability between 50 and 30 hPa is important.

But this is not the case for the Scandinavian hotspot that is climatologically strongest at 10 hPa and the fraction explained by near surface winds is lower at 50 and 30 hPa. At those levels the waves dissipate less frequently - predominantly in dependence on the stratospheric variability.

Based on the referee comment, we added following statement to the manuscript:

p7l9 This is due to the stratospheric background affecting the critical line occurrence and propagation of the GWs between 50 and 30 hPa in the eastern Asia region.

*8) Reading this, I was wondering how well the CMAM-sd simulation itself captures the different climatological patterns (NAO, SO, QBO)?*

While there are some differences between the time series of CMAM-based and observational counterparts, their similarity is generally strong, and the Pearson correlation (i.e., a correlation measure best suited for quantifying the links explored by linear regression analysis) between them is high. This is illustrated by Fig. S1 in the Supplement.

*9) This also triggers the question of how well the CMAM-sd simulation reflects reality?*

As with all model-based frameworks, the issue of reliable reproduction of the observed climate is quite complex and many statistics can be considered for validation. Our tests concentrated primarily on the ability of CMAM to reproduce the spatial patterns of response of lower tropospheric characteristics to the phases of the oscillation considered in our analysis; the results suggested high degree of match between the CMAM-based and observational data (see Fig. S2 in the supplement for a sample of the results, summarizing the 850 hPa temperature response to SO and NAO).

*10) Recent studies showed that there might be notable gravity wave activity over remote islands in the Southern Ocean, e.g., South Georgia or Kerguelen Islands. Is the CMAM-sd simulation capable of capturing this?*

From a closer look at the Figure 1, 50 hPa, DJF we can see that there is some moderately strong significant OGWD over those remote islands. However, we must note that those are monthly mean values only, not reflecting the intermittent nature of the GWs. Our study is focused globally, but the fact that CMAM is able to get some pronounced OGWD above such small islands is good news for possible future research of individual hotspots.

- Alexander, M. J., and A. W. Grimsdell (2013), Seasonal cycle of orographic gravity wave occurrence above small islands in the Southern Hemisphere: Implications for effects on the general circulation, J. Geophys. Res. Atmos., 118, 11,589–11,599, doi:10.1002/2013JD020526.

- Hoffmann, L., Grimsdell, A. W., and Alexander, M. J.: Stratospheric gravity waves at Southern Hemisphere orographic hotspots: 2003–2014 AIRS/Aqua observations, At- mos. Chem. Phys., 16, 9381-9397, https://doi.org/10.5194/acp-16-9381-2016, 2016.

**Referee #2**

**Major comments:**

1) *The author solely relies on the orographic gravity wave drag parameterization scheme of the CMAM model to examine the inter-annual variability of the orographic gravity wave drag. However, the tuning procedure as mentioned in the research may potentially overestimate the role of orographic gravity wave as compared to non-orographic source such as convections. How does the choice of tuning parameters in the orographic gravity wave drag scheme affect the conclusions in this research?*

Thank you for this comment; it is a very good point. The settings of the OGWD and NOGWD parameterization scheme used in the CMAM-sd simulation are discussed in details in McLandress et al. (2013) and we added a paragraph in the revised manuscript to make a clear summary discussing limitations connected to our results.

The tuning of OGWD consists of arbitrary choosing a value of dimensionless parameters controlling the total value of launch momentum and the vertical flux of horizontal momentum. As an indirect effect the breaking level of the waves is influenced by this setting. The nudging procedure (its influence is discussed in more detail in reply to REF#1) helps to reach realistic distributions of momentum fluxes but the breaking levels and OGWD value are largely influenced by this arbitrary tuning.

As for the overestimation of the OGW relative to NOGW role, we entirely agree and it is certain that at the analyzed levels the overestimation is present. We stressed this out more in the revised version together with statement that our conclusions are directly applicable at the model atmosphere only and thus having indirect implications for the real atmosphere.

Let us follow with an excerpt from the reply to REF#1: It would be very interesting to look at NOGWD variations connected with variability of jets, fronts etc. However, the CMAM NOGWD scheme (Scinocca, 2003) is based on launching a globally uniform isotropic NOGW spectrum in four cardinal horizontal directions at approximately 125 hPa. The aim is to produce reasonable seasonal evolution of the zonal mean zonal temperature and winds in the mesosphere and the zonal and meridional asymmetry stems from propagation effects only. Regarding NOGWD, we have produced the same analysis as for the OGWD but due to the above mentioned reasons the resulting fields are highly zonally symmetrical and weaker in magnitude compared to the OGWD and so we decided not to show them in the manuscript. However, we attach a selected figure to the manuscript supplement.

We expect the NOGWD interannual variability in the upper troposphere-lower stratosphere region to show highly zonally asymmetric behavior (storm track shifts, distribution of convection, etc.) and in our future research we would like to analyze a dataset that would at least roughly capture this.

2) *The QBO in this simulation is potentially affected by nudging. How is QBO represented in the simulation as compared to the observation? And how sensitive is the relationship between orographic gravity wave drag and QBO to nudging?*

The QBO representation is close to reality due to the nudging (specifically, the Pearson correlation of the CMAM-based and observational QBO index is 0.97 over the 1979-2010 period – see also Fig. S1 in the supplement and response to REF#1, comment 8) and

therefore also the modulation of the OGWD by QBO should be realistic. OGWD is very sensitive to the QBO mainly due to modulation of the background for the GW propagation. In our results we can also see some QBO influence on OGWD sourcing - probably due to polar vortex teleconnection. It would be highly interesting to look at the QBO influence on NOGW sourcing (convection etc.), but, as discussed above, we are not able to assess this in CMAM.

**Minor comments:**

*1) The wind vectors in figure 1, 4 and 5 are too thin to see, it help vision if the wind vectors are drawn thicker.*

Figure 1 has been modified for better readability: Thickness of the vectors has been increased, their length was converted to a nonlinear scale to better reflect high range of mean values for the gravity wave drag (similarly to the original Figs. 4 and 5) and the resolution of the bitmap components has been increased, to allow for a closer look at the details.

Figs 4, 5 and 11: Thickness and length of the vectors have been increased, as well as the resolution of the bitmap components.

*2) In figure 2, it would be more concise if the standard deviation of the wind vector amplitude (norm of the wind vector) due to orographic gravity drag rather than both zonal and meridional components are shown. This is also the case for Figure 3.*

Since variability of individual gravity wave drag components is briefly discussed in the text, we would prefer to not merge the variability components. However, to better illustrate the geographic patterns of OGWD variability, data from the original Figs. 2 and 3 has been merged into a new figure (Fig. 2 in the revised version), presenting the standard deviations in a vector-like form, thus allowing for a quick inspection of their combined magnitude.

*3) In Figure 8, 9 and 10, the author mentioned the fractions are explained by "both component", does this mean the combined norm variance of the 850-hPa wind vector?*

The value in question represents a coefficient of determination ($R^2$) associated with the multiple regression mapping using both components of 850 hPa wind (eastward and northward) as predictors; the captions of the figures have been updated to better explain this.

*4) Acronyms such as PW, SSW, IGW needs clarification.*

The acronyms are correctly introduced in the revised version of the paper.

[revised manuscript text omitted]

    **Figure S2** gives illustration of how well the CMAM-sd simulation reflects spatial relations observed in real climate system. As with all model-based frameworks, the issue of reliable reproduction of the observed climate is quite complex and many statistics can be considered for validation. Our tests concentrated primarily on the ability of CMAM to reproduce the spatial patterns of response of lower tropospheric characteristics to the phases of the oscillation considered in our analysis; the results suggested high degree of match between the CMAM-based and observational data (the sample of our results in Fig. S2 summarizes the 850 hPa temperature response to SO and NAO).

    **Figure S3** illustrates response of the non-orographic GWD. Although it would be very interesting to look at NOGWD variations connected with variability of jets, fronts etc, the CMAM NOGWD scheme (Scinocca, 2003) is based on launching a globally uniform isotropic NOGW spectrum in four cardinal horizontal directions at approximately 125 hPa. The aim is to produce reasonable seasonal evolution of the zonal mean zonal temperature and winds in the mesosphere and the zonal and meridional asymmetry stems from propagation effects only. Regarding NOGWD, we have produced the same analysis as for the OGWD but due to the above mentioned reasons the resulting fields are highly zonally symmetrical and weaker in magnitude compared to the OGWD. We expect the NOGWD interannual variability in the upper troposphere-lower stratosphere region to show highly zonally asymmetric behavior (storm track shifts, distribution of convection, etc.) and in our future research we would like to analyze a dataset that would at least roughly capture this.

[Figure]

**Figure S1.** Indices of Southern Oscillation and Quasi Biennial Oscillation, derived from CMAM data (red line) and from direct observations (blue line; observational indices provided by Climate Research Unit at https://crudata.uea.ac.uk/cru/data/soi/ and NOAA at https://www.esrl.noaa.gov/psd/data/correlation/qbo.data).

[Figure]

**Figure S2.** 850 hPa temperature response (°C) to Southern Oscillation (left) and North Atlantic Oscillation (right) change from highly negative to highly positive phase (increase of the respective index by 4x its standard deviation), evaluated through multiple linear regression. Dots represent locations with response statistically significant at the 95% level (moving-block bootstrap). For comparison with results obtained for near-ground temperature in various observational datasets see Miksovsky et al. (2016; Earth System Dynamics 7: 231-249, DOI:10.5194/esd-7-231-2016).

[Figure]

**Figure S3.** Response of the non-orographic GWD [m/s/s] at the 50 hPa level related to the activity of the Southern Oscillation (left), North Atlantic Oscillation (right) and Quasi-Biennial Oscillation (bottom); responses statistically significant at the 95% level shown in red.